# Cycle to Clique (Cy2C) Graph Neural Network: A Sight to See beyond Neighborhood Aggregation

**Yun Young Choi** [*], **Sun Woo Park** [*] **& Youngho Woo** [†]
Division of Industrial Mathematics
National Institute for Mathematical Sciences
Daejeon, South Korea

**U Jin Choi**
Department of Mathematical Sciences
KAIST
Daejeon, South Korea

## Abstract

Graph neural networks have been successfully adapted for learning vector representations of graphs through various neighborhood aggregation schemes. Previous researches suggest, however, that they possess limitations in incorporating key non-Euclidean topological properties of graphs. This paper mathematically identifies the caliber of graph neural networks in classifying isomorphism classes of graphs with continuous node attributes up to their local topological properties. In light of these observations, we construct the Cycle to Clique graph neural network, a novel yet simple algorithm which topologically enriches the input data of conventional graph neural networks while preserving their architectural components. This method theoretically outperforms conventional graph neural networks in classifying isomorphism classes of graphs while ensuring comparable time complexity in representing random graphs. Empirical results further support that the novel algorithm produces comparable or enhanced results in classifying benchmark graph data sets compared to contemporary variants of graph neural networks.

## 1 Introduction

Graph neural networks (GNN) are prominent deep learning methods for learning vector representation of graphs. Research in GNNs explores their empirical capabilities and effectiveness in classifying node labels, classifying graphs, and predicting links by modifying the message passing layers or pooling methods. These experiments support that GNNs can achieve state-of-the-art performances in executing these tasks and ensure equivalent performance to that of the Weisfeiler-Lehman (WL) isomorphism test in representing graphs with discrete node labels Xu et al. (2018). However, they have limited capabilities in incorporating global topological properties of graphs, thereby exhibiting restricted discriminative power in distinguishing isomorphism classes of graphs Bouritsas et al. (2022); Rieck et al. (2019).

To overcome these limitations, this paper presents a mathematical framework that examines which topological properties of graphs with continuous node attribute that conventional GNNs can encapsulate. Inspired by the works of Krebs and Verbitsky Krebs & Verbitsky (2015) and Xu et al Xu et al. (2018), we use the theory of covering spaces to prove that under some constraints, a pair of graphs with continuous node attributes is distinguishable by GNNs if and only if there exist isomorphisms among the collection of their finite depth unfolding trees that induce equality of induced node attributes. This gives a universal formulation which pinpoints the discriminative power of a wide range of variants of GNNs and the topological enrichments these models endow over the graph data set. Such approaches include enriching node attributes, using persistent homological techniques, gluing high dimensional complexes, and keeping track of recurring subgraph structures. Among these candidates, we focus on the procedure of transforming the cycle bases of graphs to complete subgraphs or cliques. This operation can be easily implemented by adding suitable edges to transform a cyclic subgraph into a clique and masking any other edges not included in the subgraph. The

---

[*]Equal Contribution
[†]Correspondence to: youngw@nims.re.kr

adjacency matrices obtained from the induced cliques, denoted as clique adjacency matrices, allow GNNs to effectively process the bases of cycles, which are topological properties equivalent to the first homological invariants of graphs Paton (1969). In particular, the operation can be thought as a straightforward pre-processing procedure independent from training dynamical filtration functions or attaching higher dimensional cells Horn et al. (2021); Bouritsas et al. (2022); Bodnar et al. (2021b;a), which are previously carefully studied methods for encapsulating the cyclic structures of graphs.

We thus propose the Cycle-to-Clique Graph Neural Network (Cy2C-GNN), a graph neural network whose message passing layers compute two types of hidden node attributes, each obtained from the adjacency matrix and the induced clique adjacency matrix of a graph. We confirm that Cy2C-GNN effectively processes cycle bases of graphs, thus surpassing the strengths of conventional GNNs. Experimental results support that Cy2C-GNN ensures comparable performance to GNNs that utilize persistent homological techniques with both fixed and arbitrary filtration functions. Furthermore, the simplicity of the architecture guarantees equivalent computational complexity to conventional GNNs in representing random graphs and the effective utilization of trainable parameters.

Our main contributions can therefore be summarized as follows:

1. **Theoretical Foundation:** We use the theory of covering spaces to prove that conventional GNNs fail to effectively represent cyclic structures of graphs with continuous node attributes. (Theorem 3.3, Section 3)

2. **A Simple yet Novel Network:** We propose a novel algorithm called "Cy2C-GNN" which overcomes the theoretical limitations by enriching the topological properties of the input data admitted by GNNs with clique adjacency matrices, which does not require training filtration functions or attaching high-dimensional complexes. (Theorem 4.3, Section 4)

3. **Efficient Enhancements:** The proposed algorithm effectively incorporates cyclic structures of graph data sets while ensuring equivalent computational complexity to conventional GNNs for representing random graphs and adaptability to variants of GNNs. (Section 5)

## 2 RELATED WORKS

**Graph Neural Networks (GNNs)** We recall the construction of GNNs as suggested in Xu et al Xu et al. (2018). Denote by $\text{GNN}^l$ the conventional GNN (GNN) comprised of composition of $l$ neighborhood aggregating layers. Each $m$-th layer $H^{(m)}$ of the network constructs hidden node attributes of dimension $k_m$, denoted as $h_v^{(m)}$, using the following composition of functions:

$$\begin{cases} h_v^{(m)} := \text{COMBINE}^{(m)} \left( h_v^{(m-1)}, \text{AGGREGATE}_v^{(m)} \left( \left\{\!\!\left\{ h_u^{(m-1)} \mid u \in N(v) \right\}\!\!\right\} \right) \right) \\ h_v^{(0)} := X_v \end{cases} \tag{1}$$

Here, $X_v$ is the initial node attribute at $v$, $N(v)$ is the set of nodes adjacent to $v \in V(G)$, $\text{AGGREGATE}_v^{(m)}$ is a function which aggregates features of nodes adjacency to $v$, and $\text{COMBINE}^{(m)}$ is a function which combines features of the node $v$ with those of nodes adjacent to $v$.

Denote by $H^{(l)}$ the final layer of the network. The $K$-dimensional vector representation of $G$, denoted as $h_G$, is given by

$$h_G := \text{READOUT}^{(l)} \left( \{\!\!\{ h_v^{(l)} \mid v \in V(G) \}\!\!\} \right) \tag{2}$$

where $\text{READOUT}^{(l)}$ is the graph readout function of node features updated from $l$ hidden layers. We refer readers to Appendix A.2 for a rigorous definition of graph neural networks.

A wide range of GNNs and graph representation techniques can be formulated in terms of the construction outlined above. For example, the WL test is a classical technique which consists of combining adjacent discrete node labels, substituting newly obtained labels, and constructing a complete histogram of updated labels. The test is equivalent to conventional GNNs whose aggregation and combination functions correspond to sums of multisets, and the graph readout function corresponds to a hashing function of discrete node labels. Other well-known networks whose architecture can be formulated using conventional GNNs from Section 2 include graph convolutional networks (GCN), graph attention networks (GAT), and graph isomorphism networks (GIN) Kipf & Welling (2016); Veličković et al. (2017); Xu et al. (2018).

**Covering spaces**    A number of studies carefully analyzed the strength of graph neural networks in distinguishing isomorphism classes of graphs with discrete node labels. The work by Krebs and Verbitsky Krebs & Verbitsky (2015) shows that Weisfeiler-Lehman test can distinguish a pair of graphs with constant node labels up to isomorphism of their universal covers, i.e. isomorphism of collections of finite depth unfolding trees. We refer to Appendix A.1. for a rigorous treatment of the definition of covering spaces of graphs Hatcher (2002); Krebs & Verbitsky (2015).

**Definition 2.1.** Let $G := (V, E)$ be a directed graph. For each node $v \in V(G)$, the depth-1 unfolding tree rooted at $v$, denoted as $T_v^1$, is a subtree of $G$ whose set of nodes consists of the distinguished node $v$ itself and the nodes $w$ such that there exists a directed edge from $v$ to $w$. The set of edges of $T_v^1$ are comprised of directed edges from $v$. (See Definition A.11 for a rigorous definition.)

For any positive number $k$, the depth-k unfolding tree rooted at $v$, denoted as $T_v^k$, is a subtree of $G$ whose set of nodes consists of the distinguished node $v$ itself and the nodes $w$ such that there exists at most $k$ consecutive directed edges from $v$ to $w$. The set of edges of $T_v^k$ are comprised of unions of all $k$ consecutive directed edges from $v$. (For any undirected graph, the finite depth unfolding tree rooted at $v$ is defined in an analogous manner, though the set of edges consists of undirected edges instead of directed edges). The collections of all unfolding trees of $G$ of arbitrary depth can be represented as a single graph, called the universal cover of $G$.

**Definition 2.2.** Let $G := (V, E)$ be a connected graph. The universal cover of $G$ is a connected tree $\tilde{G}$ (possibly infinite) with the projection map $\pi_G : \tilde{G} \to G$ that maps any depth-1 unfolding trees in $\tilde{G}$ isomorphically to some depth-1 unfolding tree in $G$. (See Definitions A.12 for more details.)

For graphs with several connected components, their universal covers are disjoint unions of connected trees, each component of which satisfies the local isomorphism condition from Definition 2.2. We recall that Xu et al. proved that GNNs with injective aggregation, combination, and graph readout functions share equivalent strengths with Weisfeiler-Lehman test in distinguishing non-isomorphism classes of graphs with discrete node labels Xu et al. (2018). Combined with Krebs and Verbitsky's result, we obtain that such GNNs can distinguish pairs of graphs with constant node labels up to isomorphism classes of their universal covers, or finite depth unfolding trees.

**Improvements**    Recent studies focused on excavating novel techniques which may outperform or optimize GNNs in analyzing graph data sets, such as classifying node attributes, classifying graph datasets, and predicting links among the set of nodes. Graph kernels measure similarities among graphs by utilizing inner products between their representations, a form of an embedding to a reproducing kernel Hilbert space. Borgwardt et al. (2020); Kashima et al. (2003); Shervashidze et al. (2011); Vishwanathan et al. (2010) Persistent homological techniques and topological data analytic tools have garnered attention as key sources of encapsulating global topological invariants of graphs, such as counting the number of connected components or cycles present in given graph datasets. Several studies focused on incorporating these topological tools with graph kernels by using pre-defined filtration functions Rieck et al. (2019), or with GNNs by constructing task-specific filtration functions Hofer et al. (2020); Horn et al. (2021). Keeping track of topological substructures of graphs is also shown to be effective in enriching the quality of vector representations obtained from GNNs Rieck et al. (2017); Sizemore et al. (2017). Other approaches focus on proposing novel message passing schemes as measures to address the limited performance of conventional GNNs Bodnar et al. (2021b); Bouritsas et al. (2022).

## 3    GNNs AND UNIVERSAL COVERS

In this section, we utilize the theory of covering spaces to assess the quality of representations conventional GNNs construct from collections of graphs with continuous node attributes.

**Definition 3.1.** Let $G := (V, E)$ be a graph. Denote by $G' := (V, E')$ the graph where every node has a self-loop. Denote by $G'' := (V, E'')$ the directed graph with a projection map $p : G'' \to G'$ such that each undirected edge of $G'$ from $v_1$ to $v_2$ is constructed as follows: If $v_1 \neq v_2$, the undirected edge corresponds to two directed edges, one edge from $v_1$ to $v_2$, and the other from $v_2$ to $v_1$: Otherwise, the edge corresponds to a single directed self-loop from $v_1$ to itself.

Figure 1 illustrates the construction of $G'$ and $G''$. Recall that the graph $G$ is endowed with continuous attributes $f_G : V(G) \to \mathbb{R}^k$. Respecting the configuration of these attributes, we can endow new

node attributes to $G''$ and the universal cover $\tilde{G}''$. We call these new assignments of node attributes to the universal cover $\tilde{G}''$ the pullback of node attributes.

**Definition 3.2.** Let $f_G : V(G) \to \mathbb{R}^k$ be the function of node attributes over the graph $G$. The pullback of the node attributes to the associated universal cover $\tilde{G}''$ is the composition of functions $f_{G''} \circ \pi_{G''} : V(\tilde{G}'') \to \mathbb{R}^k$, where $f_{G''} : V(G'') \to \mathbb{R}^k$ is the node attributes on $G''$ obtained from using the identical node attributes on $G$. (See Definitions A.10 for a rigorous formulation.)

Inspired from the effectiveness of Weisfeiler-Lehman isomorphism tests Krebs & Verbitsky (2015) and GNNs Xu et al. (2018), we prove that GNNs with injective message passing layers represent a pair of graphs with continuous node attributes as distinct vectors if and only if there exists an isomorphism between their associated universal covers whose node attributes endowed in a manner that respects the node attributes of the original graphs are identical, i.e. there exist isomorphisms among the collection of their finite depth unfolding trees that induce equality of the pullback of node attributes. We refer the proof of the theorem to Appendices A.3. and A.4.

**Theorem 3.3.** *Let $\mathcal{G}$ be a collection of finite undirected connected graphs with continuous node attributes. Let $G, H \in \mathcal{G}$ be any two connected graphs with the same number of nodes. Denote by $f_G : V(G) \to \mathbb{R}^k$ and $f_H : V(H) \to \mathbb{R}^k$ an arbitrary choice of continuous node attributes. Let $d_G, d_H$ be the diameters of graphs $G$ and $H$, i.e. the maximum of the shortest length of paths between any two nodes (see Appendix A.4 for the rigorous definition.)*

*Suppose a graph neural network $GNN^l$ with $l$ layers satisfies the following three constraints: (1) For every $m$ such that $1 \leq m \leq l$ and for each $v \in V(G)$, the functions $AGGREGATE_v^{(m)}$ and $COMBINE^{(m)}$ are injective: (2) The graph read-out function READOUT is injective: And (3) $l \geq 2\max(d_G, d_H)$. Then $GNN^l$ maps the pair of graphs $G, H \in \mathcal{G}$ to identical vector representations if and only if there exists an isomorphism of universal covers $\varphi : \tilde{G}'' \to \tilde{H}''$ whose node attributes respecting the node attributes over $G$ and $H$ are identical, i.e.*

$$h_G = h_H \iff \exists \, isomorphism \, \varphi : \tilde{G}'' \to \tilde{H}'' s.t. \, f_{G''} \circ \pi_{G''} = f_{H''} \circ \pi_{H''} \circ \varphi \qquad (3)$$

*Remark* 3.4. The above theorem outlines the maximal extent of GNNs in detecting non-isomorphism classes of graphs, a generalization of results proved in Krebs & Verbitsky (2015), Xu et al. (2018), and a contemporary result from Bamberger (2022). It also pinpoints that conventional GNNs whose number of layers are at least twice the maximum diameter of a graph data set are sufficient to fully distinguish isomorphism classes of graphs up to their universal covers, which to the best of our knowledge enhances the results from previous researches that focused on analyzing the performance of GNNs with sufficiently large numbers of layers. Meanwhile, universal covers of graphs are infinite trees that do not contain any cycles, see Chapter 1.3 of Hatcher (2002) for example. Thus, conventional GNNs, regardless of the injectivities of the three functions, have limited capability in incorporating homological invariants of graphs, such as cyclic subgraph structures of a finite graph.

## 4 CYCLE-TO-CLIQUE GRAPH NEURAL NETWORKS (CY2C-GNN)

**Motivation** As shown in Theorem 3.3, GNNs can distinguish a collection of finite graphs up to isomorphism classes of their universal covers and equivalences of pullbacks of node attributes, but may fail to distinguish a collection of graphs with isomorphic universal covers whose subgraphs are comprised of cyclic graphs with different number of nodes, see Figure 1 for instance. We note that Theorem 3.3 proves a rigorous mathematical equivalence relation between the conditions conventional GNNs satisfy and the nature of isomorphism classes of graphs conventional GNNs can distinguish. Thus, any novel GNN model which twists one of the conditions from Theorem 3.3 can be considered as a candidate model for outperforming conventional GNNs.

Using Theorem 3.3, we deduce four possible approaches which may enrich the input data structure of graphs the GNN may process to distinguish non-isomorphic classes of graphs. **(1) Assignment of unique node attributes**: One can assign new attributes to each node. Practical implementations include assigning node degrees as labels, imposing positional encoding, implementing labeling tricks, inductive identity coloring, and random assignment of node features. Dwivedi et al. (2022); Geerts et al. (2021); You et al. (2019); Zhang et al. (2021); You et al. (2021); Abboud et al. (2021). **(2) Persistent homological methods**: We may incorporate homological invariants of graphs to GNNs

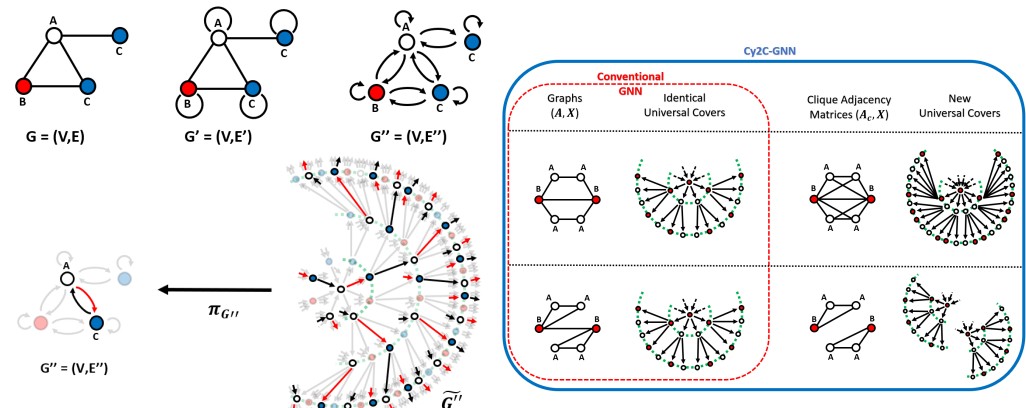

Figure 1: **(Upper Left)** An exemplary graph $G$ with the induced graphs $G'$ and $G''$. **(Lower Left)** An illustration of pullback of node attributes (Definition 3.2) defined over the base graph $G''$ to its universal cover $\tilde{G}''$. The corresponding node attributes and edges over the two graphs are marked in identical colors. **(Right)** An illustration of Theorem 3.3 and Theorem 4.3. The two graphs $G$ and $H$ have identical associated universal covers $\tilde{G}''$ and $\tilde{H}''$ and equality of pullback of node attributes.

by utilizing or constructing height functions over nodes and edges and constructing the associated persistence diagrams Carrière et al. (2020); Hofer et al. (2020); Horn et al. (2021); Rieck et al. (2019). These diagrams allow one to keep track of variations in the number of components and cycles throughout the filtration of graph dataset induced from the choice of the height functions. **(3) Topological Enrichments**: One can also enrich topological structures of graphs by attaching simplicial and cellular complexes, or incorporating subgraph structures Bodnar et al. (2021b;a); Bouritsas et al. (2022). This allows GNNs to capture topological substructures of graphs, such as cliques or cyclic subgraphs. **(4) Non-isomorphic universal covers**: We may construct a collection of new graphs with non-isomorphic universal covers induced from cyclic subgraphs. We can thus allow conventional GNNs to represent a collection of cyclic subgraphs as distinct vectors without significantly altering their architectural designs.

The last approach hints a natural procedure orthogonal to the first three previously studied approaches: Substitute cyclic graphs with complete graphs of the same number of nodes. This can be rigorously formulated as in the following lemma, whose proof is in Lemma A.30.

**Lemma 4.1.** *Denote by $C_n$ the cyclic undirected graph without self-loops consisting of $n$ nodes, and $K_n$ the undirected complete graph with $n$ nodes without self-loops. Then for any $m_1 \neq m_2$, the universal covers of cyclic graphs $\tilde{C_{m_1}}''$ and $\tilde{C_{m_2}}''$ are isomorphic, whereas the universal covers of complete graphs $\tilde{K_{m_1}}''$ and $\tilde{K_{m_2}}''$ are not.*

We thus devise a GNN which admits both the adjacency matrix of a graph $G$ and the adjacency matrix of unions of cliques induced from substituting cyclic subgraphs with complete subgraphs.

**Definition 4.2** (Clique Adjacency Matrix). Let $G := (V, E)$ be an undirected graph. Fix the cycle basis $\mathcal{B}_G$ of $G$, the set of cyclic subgraphs of $G$ which forms the basis of the cycle space (or the first homology group) of $G$. The clique adjacency matrix of $G$, denoted as $A_C$, is the adjacency matrix of the union of $\#\mathcal{B}_G$ complete subgraphs, each obtained from adding all possible edges among the set of nodes of each basis element $B \in \mathcal{B}_G$. Note that such a $B$ is a cyclic subgraph of $G$. Explicitly, the matrix $A_C := \{a_{u,v}^C\}_{u,v \in V(G)}$ is given by

$$a_{u,v}^C := \begin{cases} 1 & \text{if } \exists\, B \in \mathcal{B}_G \text{ cyclic s.t. } u, v \in V(B) \\ 0 & \text{otherwise} \end{cases} \tag{4}$$

Given $3 \leq c_1 \leq c_2 < \infty$, one may also define the bounded clique adjacency matrix $A_C|_{[c_1, c_2]} := \{a_{u,v}^C|_{[c_1, c_2]}\}_{u,v \in V(G)}$ which only substitutes cycles of size between $c_1$ and $c_2$ to cliques:

$$a_{u,v}^C|_{[c_1, c_2]} := \begin{cases} 1 & \text{if } \exists\, B \in \mathcal{B}_G \text{ cyclic s.t. } u, v \in V(B),\ c_1 \leq |V(B)| \leq c_2 \\ 0 & \text{otherwise} \end{cases} \tag{5}$$

For each node $v \in V(G)$, we denote by $\mathcal{C}(v)$ the set of nodes $w \in V(G)$ such that there exists a cyclic subgraph $C \in \mathcal{B}_G$ such that both $v$ and $w$ lie in $C$:

$$\mathcal{C}(v) := \{w \in V(G) \mid \exists\, B \in \mathcal{B}_G \text{ s.t. } v, w \in V(B)\} \tag{6}$$

Required for computing the set $\mathcal{C}(v)$ for each node $v \in V(G)$, is the basis of cycles of the graph $G$. Here, the cycle basis, denoted as $\mathcal{B}_G$, refers to the minimal set of cyclic subgraphs of $G$ whose combinations generate all possible cyclic subgraphs. To construct the basis, we use the algorithm proposed by Paton, whose time complexity for computing the basis for a graph with $n$ nodes and $m$ edges is of order $O(n^\gamma)$ for $2 \leq \gamma \leq 3$, and that for a random graph with $n$ nodes is of order $O(n^2)$ Paton (1969). Given a predetermined ordering on $E(G)$, we can use persistent homological techniques to further reduce the time complexity required for obtaining $\mathcal{B}_G$ to $O(m\alpha(m))$, where $\alpha(m)$ is the inverse Ackermann function which can be regarded as a constant for any practical values of $m$ Horn et al. (2021). In other words, the best time complexity required for preprocessing the given graph data set is of order practically equivalent to $O(m)$ where $m$ is the number of edges of $G$.

**Cycle-to-Clique Graph Neural Network**     Using the clique adjacency matrices, we now propose the Cycle-to-Clique Graph Neural Network (Cy2C-GNN), a novel and simple modification of GNNs ensuring low computational complexity and efficient usage of trainable weights. The topological features of a graph $G := (V, E)$ with continuous node attributes $f_G : V(G) \to \mathbb{R}^k$ can be represented by three types of matrices: The adjacency matrix $A \in \mathbb{R}^{n \times n}$: The node feature matrix $X \in \mathbb{R}^{n \times k}$ obtained from the function $f_G$: And the clique adjacency matrix $A_C \in \mathbb{R}^{n \times n}$ (possibly bounded) from Definition 4.2. The Cy2C-GNN admits the three types of matrices $(A, A_C, X)$ as inputs, whereas conventional GNNs only utilize two types of matrices $(A, X)$.

The Cy2C-GNN consists of two sets of neighborhood aggregating layers: A single layer utilizing the clique adjacency matrix $A_C$: And $l$ neighborhood aggregating layers identical to conventional GNN layers, as defined in Section 2, which utilize the usual adjacency matrix $A$. The model thus preserves various executive merits of GNNs, such as incorporation of edge attributes and efficient applicability to large graph datasets. The trainable weights are not shared among the hidden layers.

The first layer of Cy2C-GNN is a single neighborhood aggregating layer utilizing the clique adjacency matrix $A_C \in \mathbb{R}^{n \times n}$. The output of the first hidden layer is the hidden attribute $c_v^{(1)}$ obtained from the usual aggregation and combination functions used to obtain hidden vectors of conventional GNNs:

$$c_v^{(1)} = \text{COMBINE}^{(1)}(X_v, \text{AGGREGATE}_v^{(1)}(\{\!\{X_u : u \in \mathcal{C}(v)\}\!\})) \tag{7}$$

Recall that $\mathcal{C}(v)$ is a set of nodes which are adjacent to $v$ in the clique adjacency matrix. Here, the functions $\text{AGGREGATE}_v^{(1)}(\cdot)$ and $\text{COMBINE}^{(1)}(\cdot)$ are functions as defined in Section 2.

To entwine the local topological properties with cyclic structures of a graph, we implement a conventional GNN comprised of $l$ neighborhood aggregating layers, disjoint from the single layer utilizing clique adjacency matrices. Each $m$-th layer outputs the hidden attribute $h_v^{(m)}$ which is inductively defined as follows.

$$\begin{cases} h_v^{(m)} := \text{COMBINE}^{(m)}\left(h_v^{(m-1)}, \text{AGGREGATE}_v^{(m)}\left(\left\{\!\!\left\{h_u^{(m-1)} \mid u \in N(v)\right\}\!\!\right\}\right)\right) \\ h_v^{(0)} := X_v \quad \text{for } 1 \leq m \leq l \end{cases} \tag{8}$$

The hidden node attributes obtained from pairs of layers $c_v^{(1)}$ and $h_v^{(l)}$ are concatenated, followed by multi-layer perceptronos (MLPs) to obtain the final hidden node attribute $H_v$:

$$H_v = \text{MLP}(\text{CONCAT}(c_v^{(1)},\ h_v^{(l)})). \tag{9}$$

As for obtaining the vector representation of a graph $G$, the Cy2C-GNN separately aggregates the hidden attributes $c_v^{(1)}$ and $h_v^{(l)}$ for each node $v$, followed by a composition with MLPs.

$$\begin{aligned} H_{h_G, c_G} &= \text{MLP}(\text{CONCAT}(H_{h_G},\ H_{c_G})) \\ H_{h_G} &= \text{READOUT}^{(l)}(\{\!\{h_v^{(l)} \mid v \in V(G)\}\!\}) \\ H_{c_G} &= \text{READOUT}^{(1)}(\{\!\{c_v^{(1)} \mid v \in V(G)\}\!\}) \end{aligned} \tag{10}$$

**Discriminative Power of Cy2C-GNN**     Cy2C-GNN can distinguish a collection of unions of cyclic graphs, each comprised of possibly different number of nodes, the non-isomorphic classes of graphs of which GNNs may not distinguish. We refer to Theorem 4.3 and Figure 1 for some pairs of graphs with node attributes that Cy2C-GNN can distinguish, whereas conventional GNNs cannot.

**Theorem 4.3.** *Let $G$ and $H$ be graphs which have isomorphic universal covers, endowed with node features $f_G : V(G) \to \mathbb{R}^k$ and $f_H : V(G) \to \mathbb{R}^k$. Fix a cycle basis $\mathcal{B}_G$ of $G$. Suppose that there exists a chordless cyclic subgraph $C \in \mathcal{B}_G$ such that any cycle basis $\mathcal{B}_H$ does not have any chordless cyclic subgraph of size equal to $|V(C)|$. Then Cy2C-GNN which utilizes bounded clique adjacency matrices can distinguish $G$ and $H$, whereas conventional GNNs cannot.*

Here, a subgraph $H \subset G$ is chordless if no other cyclic subgraphs $C \in G$ satisfy $V(C) \subsetneq V(H)$. Hence, Cy2C-GNN can distinguish classes of graphs that 1,2, and 3-dimensional WL tests cannot distinguish, in particular certain classes of strongly regular graphs. (See for example Remark A.37 for further details). Theorem 3.3 and Lemma 4.1 further suggest that the first layer of Cy2C-GNN utilizing clique adjacency matrix is enough to distinguish such pairs of graphs, a marked improvement from other contemporary GNNs which assume to have sufficiently large numbers of hidden layers. We leave the proof of Theorem 4.3 to Theorem A.35 as well as comparison in discriminative power of Cy2C-GNN to other contemporary state-of-the-art GNNs to Example A.36 and Remark A.37.

**Computational Complexity**     Because the Cy2C-GNN algorithm preserves the conventional neighborhood aggregating layers of GNNs, the time complexity of representing a connected graph $G$ with $n$ nodes and $m$ edges using the Cy2C-GNN algorithm with $l + 1$ layers is equal to $O(m_C + lm)$, where $m_C$ is the number of edges of the graph associated to the clique adjacency matrix $A_C$ of $G$. By the Euler characteristic formula, the number of elements in the cycle bases $\mathcal{B}_G$ of a connected graph $G$ is equal to $m - n + 1$. Hence, for connected graphs with bounded number of nodes constituting the subgraphs of their cycle bases, Cy2C-GNN is comparable to time complexity of conventional GNNs, and more efficient than spectral decomposition of adjacency matrices of finite graphs and constructing persistence diagrams using trainable or dynamic filtration functions Milosavljeviç et al. (2011); Rieck et al. (2019). In addition, the time complexity for preprocessing the graph $G$ to obtain clique adjacency matrices is practically equivalent to $O(m)$, without requiring any training of filtration functions for each graph $G$. We refer to Appendix A.6 for a detailed discussion on the computational complexity of these algorithms.

## 5 EXPERIMENTS

**Dataset**     To analyze the effectiveness of Cy2C-GNN in distinguishing graphs with varying cyclic subgraphs, we perform an ablation study by utilizing the "CYCLES" and "NECKLACES" synthetic datasets constructed from Horn et al. (2021). These datasets are comprised of graphs containing cyclic subgraphs, which are designed to assess whether the given GNN can identify differences among such cyclic substructures. As for evaluating the effectiveness of the proposed models in classifying graph datasets, we use the 3 bioinformatics(DD, PROTEINS(FULL), ENZYMES), 3 social network datasets (COLLAB, IMDB-B, REDDIT-B ), and 3 small molecular datasets (MUTAG, NCI1, NCI109). To further verify the extendability of Cy2C-GNN models to graph datasets with additional attributes, we utilized 3 datasets with edge features (BZR-MD, COX2-MD, PTC-MR) as well. Lastly, we tested Cy2C-GNN models on 4 large datasets (REDDIT-M-5K, MOLHIV, MOLTOX21, MOLTOXCAST) from TU datasets Morris et al. (2020) and Open Graph Benchmark datasets Hu et al. (2021) to test their efficiency in processing large graph data sets in comparison to GNNs based on persistent homological techniques. The details of all the aforementioned datasets, obtained from the pytorch-geometric library, are summarized in Appendix B.

**Models**     To assess whether improvements in the discriminative power of Cy2C-GNN lead to enhancements in classifying benchmark graph datasets, we additionally implemented three baseline models comprised of Graph Convolutional Network(GCN), Graph Attention Networks(GAT), and Graph Isomorphism Network(GIN). All baseline models share the same structure with Cy2C-GNNs except for additional structures required for implementing clique adjacency matrices. We also compare the best classification results obtained from Cy2C-GNN with baseline GNNs, those of the kernel methods(WL (Borgwardt et al., 2020, Table 4.5), WL-OA (Borgwardt et al., 2020, Table 4.5), RetGK (Ye et al., 2020, Table 3), HGK (Togninalli et al., 2019, Table 2), GH (Togninalli et al., 2019, Table 2)), kernel method with persistent homology(PWL) (Rieck et al., 2019, Table

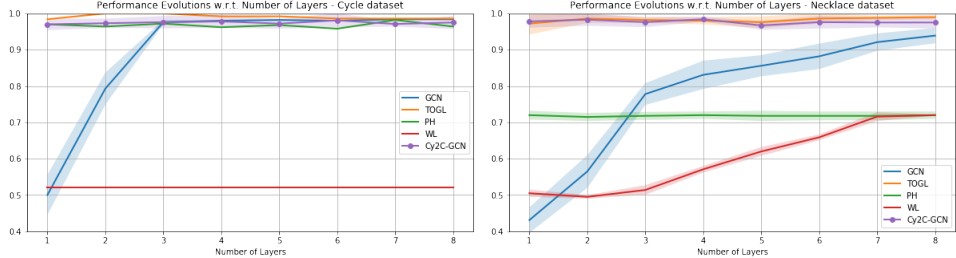

Figure 2: Comparisons of classification results obtained from baseline models and Cy2C-GNN from synthetic CYCLE **(left)** and NECKLACE **(right)** datasets Horn et al. (2021). All other four baseline models are obtained from (Horn et al., 2021, Figure 1).

1), GNN models(Gated-GCN (Dwivedi et al., 2020, Table16,seed1), GMT (Baek et al., 2021, Table 1), DGCNN (Wijesinghe & Wang, 2022, Table 14), PNA (Corso et al., 2020, Figure 6), ID-GNN (You et al., 2021, Table 3), GraphSNN (Wijesinghe & Wang, 2022, Table 3,4,14)) and GNN models with persistent homology(PersLay (Carrière et al., 2020, Table 2), TOGL (Horn et al., 2021, Table 2,3)), which aligned to the experimental protocols tested for Cy2C-GNNs. For OGB datasets(MOLHIV, MOLTOX21, MOLTOXCAST), we also consider GNNs with virtual node(VN) methods Hu et al. (2021). We omitted classification results obtained from other contemporary GNNs whose experimental procedures are different from those suggested in this paper to avoid biased comparisons of proposed GNNs, as carefully suggested in Errica et al. (2020). Further elaborations on the aforementioned models are outlined in Appendix B.1.

**Cy2C-GNN Setup**    In accordance with the experimental setup proposed by Dwivedi et al. Dwivedi et al. (2020) and Horn et al. Horn et al. (2021), we imposed that the architectural components of the baseline GCN, GAT, and GIN models, such as the number of layers, hidden attribute dimensions, and the classes of aggregation and combination functions, be identical to those of Cy2C-GNN. Differences in the number of trainable parameters among these models mostly occur at the final layer, where Cy2C-GNN harbors additional MLPs utilized for representing graphs as real vectors. For additional test on dataset with edge features and large dataset, we only consider Cy2C-GCN. We performed additional hyperparameter optimizations for implementing Cy2C-GNNs. Further details on the differences among the implemented networks are explicated in Appendix B.1.

**Ablation Studies**    Figure 2 illustrates the classification results obtained from baseline models and Cy2C-GCN from synthetic CYCLE (a) and NECKLACE (b) datasets Horn et al. (2021). We denote by "GCN" graph convolutional networks, "TOGL" topological graph neural networks which model dynamic persistent homological techniques, "PH" static persistent homological techniques, and "WL" the 1-dimensional WL test, all results of which were obtained from (Horn et al., 2021, Figure 1). The Cy2C-GCN model detects cyclic structures of graphs as effectively as persistent homological techniques, which utilize dynamic choices of graph features. We note that Cy2C-GCN can effectively distinguish cyclic graphs with number of nodes at least 4, because Lemma 4.1 implies that cyclic graphs and complete graphs of sizes 3 are isomorphic. Placing a single neighborhood aggregating layer utilizing clique adjacency matrices ahead of other conventional layers proves to be effective in detecting desired cyclic structures, as implied from Lemma 4.1.

**Results**    Comparisons among the proposed Cy2C-GNN and contemporary graph representation techniques on benchmark dataset are listed in Table 1. The Cy2C-GNN produces outperforming classification results than baseline GNNs on all of the benchmark datasets. With the exception of NCI dataset, we confirm that the Cy2C-GNNs exhibit better or similar performance to variants of WL tests and conventional GNNs among bioinformatics, social network, and small molecules datasets. Furthermore, we compare Cy2C-GNN model on dataset with edge features and large dataset to verify the robustness of Cy2C-GNN models in representing graph data sets with additional features. Cy2C-GNN shows equivalent or outperforming performance to other GNNs and persistent homological techniques in classifying most graph datasets except PTC-MR and MOLHIV datasets. These results demonstrate that Cy2C-GNN has the potential to efficiently incorporate cyclic structures of large graph datasets to message passing layers, even for datasets with edge features. We also verify that

Table 1: Classification results obtained from bioinformatics, social network and small molecules dataset. Note that N/A indicate graph classification methods which do not report classification results on the given graph data set. Classification methods with grey color text are cited from available results obtained from pre-existing publications.

| | Bioinformatics | | | Social network | | | Small molecules | | |
|---|---|---|---|---|---|---|---|---|---|
| | D&D | PROTEINS(FULL) | ENZYMES | COLLAB | IMDB-B | REDDIT-B | MUTAG | NCI1 | NCI109 |
| WL | 77.7±2.0 | 73.1±0.5 | 54.3±0.9 | 68.3±1.5 | 71.2±0.5 | 78.0±0.6 | 85.75±1.96 | 85.60±0.36 | 85.76±0.22 |
| WL-OA | 77.8±1.2 | 73.5±0.9 | 58.9±0.9 | 80.18±0.25 | 74.0±0.7 | 87.6±0.3 | 86.10±1.95 | **85.95±0.23** | **86.17±0.19** |
| PWL | 78.50±0.41 | 75.86±0.38 | N/A | N/A | N/A | N/A | 85.17±0.29 | 85.62±0.27 | 85.11±0.30 |
| Gated-GCN-4 | 72.92±2.09 | **76.36±2.90** | 65.67±4.90 | N/A | N/A | N/A | N/A | N/A | N/A |
| GMT | 78.72±0.59 | 75.09±0.59 | N/A | 80.74±0.54 | 73.48±0.76 | N/A | 83.44±1.33 | N/A | N/A |
| DGCNN | 76.6±4.3 | 72.9±3.5 | 38.9±5.7 | 71.2±1.9 | 69.2±3.0 | 49.2±1.2 | N/A | 76.4±1.7 | N/A |
| GraphSNN | 77.1±3.3 | 74.5±3.5 | 61.7±34 | 77.0±3.1 | 72.3±3.6 | 57.1±3.1 | N/A | 81.6±2.8 | N/A |
| PersLay | N/A | 74.8 | N/A | 76.4 | 71.2 | N/A | **89.8** | 73.5 | 69.5 |
| TOGL | 75.2±4.2 | 76.0±3.9 | 53.0±9.2 | N/A | N/A | N/A | 90.4±1.4 | N/A | 75.8±1.8 | N/A |
| Baseline GCN | 78.10±3.33 | 74.84±2.92 | 64.17±6.07 | 82.18±1.73 | 73.90±4.61 | 93.60±1.61 | 83.01±9.01 | 79.59±2.87 | 78.48±2.00 |
| | (GCN-1) | (GCN-3) | (GCN-3) | (GCN-4) | (GCN-5) | (GCN-4) | (GCN-5) | (GCN-5) | (GCN-4) |
| Baseline GAT | 78.10±3.49 | 74.84±3.68 | 68.33±3.73 | 80.78±1.85 | 73.50±4.99 | 92.50±1.69 | 75.53±9.77 | 78.71±2.38 | 77.03±2.42 |
| | (GAT-1) | (GAT-3) | (GAT-3) | (GAT-4) | (GAT-2) | (GAT-4) | (GAT-3) | (GAT-4) | (GAT-5) |
| Baseline GIN | 68.59±3.58 | 69.53±3.33 | 58.17±6.97 | 81.57±1.90 | 72.80±5.27 | 83.60±2.72 | 87.16±7.69 | 79.22±2.78 | 79.52±2.25 |
| | (GIN-4) | (GIN-3) | (GIN-5) | (GIN-1) | (GIN-5) | (GIN-2) | (GIN-4) | (GIN-5) | (GIN-4) |
| Cy2C-GNN | **78.86±2.22** | 76.19±4.21 | **72.83±4.60** | **83.18±1.53** | **76.40±4.41** | **94.05±1.90** | 88.89±7.57 | 80.85±2.01 | 80.78±2.03 |
| | (Cy2C-GCN-3) | (Cy2C-GCN-4) | (Cy2C-GAT-2) | (Cy2C-GCN-4) | (Cy2C-GCN-4) | (Cy2C-GCN-5) | (Cy2C-GIN-2) | (Cy2C-GIN-3) | (Cy2C-GIN-4) |

Table 2: Classification results obtained from datasets with edge features and large datasets. Note that N/A indicate graph classification methods which do not report classification results on the given graph data set. Classification methods with grey color text are cited from available results obtained from pre-existing publications.

| | datasets with edge features | | | | Large datasets | | | |
|---|---|---|---|---|---|---|---|---|
| | BZR-MD | COX2-MD | PTC-MR | | | REDDIT-M-5K | MOLHIV | MOLTOX21 | MOLTOXCAST |
| WWL | 69.76±0.94 | 76.33±1.02 | **66.31±1.21** | GIN | 56.1±1.7 | 75.58±1.4 | 74.91±0.51 | 63.41±0.74 |
| RETGK-12 | 62.77±1.69 | 59.47±1.66 | 62.5±1.6 | GIN+VN | N/A | 75.2±1.3 | 76.21±0.82 | 66.18±0.68 |
| HGK-WL | 68.94±0.65 | 74.61±1.74 | N/A | PNA | N/A | 79.05±1.3 | N/A | N/A |
| HGK-SP | 66.17±1.05 | 71.83±1.61 | N/A | ID-GNN | N/A | 78.30±2.0 | N/A | N/A |
| GH | 69.14±2.08 | 66.2±1.05 | N/A | DGCNN | 49.2±1.2 | N/A | N/A | N/A |
| GraphSNN | N/A | N/A | 61.63±2.8 | GraphSNN | **57.1±3.1** | **79.72±1.83** | 76.78±1.27 | **67.68±0.92** |
| Cy2C-GCN | 72.60±7.77 | 77.22±7.09 | 64.28±9.83 | **Cy2C-GCN** | 57.03±1.9 | 78.02±0.6 | **77.34±0.68** | 67.61±0.21 |
| (Best) | (Cy2C-GCN-5) | (Cy2C-GCN-3) | (Cy2C-GCN-5) | (Best) | (Cy2C-GCN-5) | (Cy2C-GCN-5) | (Cy2C-GCN-7) | (Cy2C-GCN-3) |

Cy2C-GNNs are sufficient enough to incorporate topological structures while managing to reduce computational costs compared to dynamic persistent homological techniques, which require users to find suitable filtration functions for each given graph dataset.

## 6 CONCLUSION

In this paper, we utilized the theory of covering spaces to formulate a mathematical framework entailing the strengths of conventional GNNs in detecting isomorphism classes of graphs with continuous node attributes. These mathematical observations lead us to propose Cy2C-GNN, which enriches the topological characteristic of input data by utilizing clique adjacency matrices. We demonstrated both theoretically and experimentally that the proposed network can efficiently and reliably represent cyclic (or homological) structures of graph data sets without undergoing major alterations in the architectural structure of conventional message passing layers, such as training dynamic filtration functions or gluing high-dimensional cells or complexes.

Nevertheless, Cy2C-GNN is not a panacea for resolving the problem of distinguishing all non-isomorphic classes of graphs. For instance, Cy2C-GNN does not guarantee to distinguish collections of graphs whose number of nodes of all cyclic subgraphs are equal to each other. Precautionary measures are also required in stacking a large number of neighborhood aggregating layers utilizing clique adjacency matrices, as it may increase the risk of oversmoothing the node features from significantly decreasing the graph diameters. These risks must be taken into account when applying Cy2C-GNN layers for classifying node labels or performing graph regression tasks. We hence advise to use a single layer of Cy2C-GNN preceding conventional GNNs, which is enough to discern graphs with markedly different cyclic subgraph structures. Future research may focus on identifying the extent of how much homological structures Cy2C-GNN can incorporate, identifying It would be of great interest to analyze the inherent relations among Cy2C-GNN and other variants of GNNs, and experiment whether combining Cy2C-GNNs with other state-of-the-art techniques may further enhance their performances.

## ACKNOWLEDGEMENT

This research was supported by the National Institute for Mathematical Sciences (NIMS) grant funded by the Korean Government (MSIT)(No.B23910000)

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

# A    PROOFS

## A.1    CELL COMPLEXES AND COVERING SPACES

In this section, we utilize the theory of covering spaces to provide a rigorous formulation that Weisfeiler-Lehman test encapsulates local topological properties of graphs by representing finite depth unfolding trees using node attributes. Readers who are interested in a rigorous treatment of the theory of covering spaces may refer to Hatcher (2002) or Krebs & Verbitsky (2015). Krebs and Verbitsky Krebs & Verbitsky (2015) proved that the Weisfeiler-Lehman test represents a pair of graphs with fixed constant node labels as identical vectors if and only if their universal covers are isomorphic. Combined with the results on the equivalence between graph neural networks and Weisfeiler-Lehman tests Xu et al. (2018), we prove that graph neural networks represents a pair of graphs with arbitrary node labels as identical vectors if and only if there exists a graph isomorphism between their universal covers that induces an equality of node labels on the universal covers.

Throughout this paper, we consider a graph $G := (V, E)$ as a cell complex, which is constructed as follows.

**Definition A.1** (Graph as a cell complex). A graph $G := (V, E)$ may be constructed in the following procedure.

1. The set of nodes $V := V(G)$ corresponds to a discrete set of points $\{v\}_{v \in V(G)}$ (0-cells).

2. The set of edges $E := E(G)$ corresponds to a discrete set of intervals $\{[0, 1]\}_{e \in E(G)}$ (1-cells).

3. The graph $G$ is inductively constructed by attaching the endpoints of the edges to their corresponding nodes. That is, given an interval $[0, 1]$ corresponding to an edge $e := (v_1, v_2)$, one glues the endpoint $\{0\}$ to the node $v_1$, and the endpoint $\{1\}$ to the node $v_2$.

4. One may iterate the inductive process finitely or indefinitely many times, depending on the cardinality of the set of edges.

We call the spaces constructed in this manner as a 1-dimensional cell complex.

Cell complexes allow one to endow graphs with topological structures induced from that over the set of discrete points and the unit interval $[0, 1]$. Let us recall that the subset of points characterizes the topological structure defined over the discrete set of points, whereas the open intervals $(a, b)$ with $0 \le a \le b \le 1$ characterizes the topological structure defined over $[0, 1]$. In an analogous manner, the open subsets characterizing the topological structure over graphs are the subset of nodes, the open intervals defined over an edge, and the open subset centered at a node $v$, obtained from gluing a set of open sub-intervals with common endpoints $v$. Taking countable unions and finite intersections of these three types of open subsets, one shall construct any subsets which represent local topological properties of $G$.

**Definition A.2** (Covering Space). Let $G$ be a graph. Let $x \in G$ be a point, which could be a node $v \in V(G)$ or any point on an edge $e \in E(G)$. A covering space of $G$ is a graph $\hat{G}$ with a projection map $p_G : \hat{G} \to G$ such that for any point $x \in G$, there exists an open subset $U \subset G$ whose pre-image $p_G^{-1}(U)$ is a disjoint union of open subsets $\{U_i\}$, each of which are homeomorphic to $U$.

In other words, a covering space $\hat{G}$ of $G$ is a graph whose local topological properties are equivalent to those of $G$. In this paper, we will consider the universal cover of $G$, which is a canonical graph whose subgraphs correspond to paths in $G$ starting at a node $v \in V(G)$ up to homotopy, an equivalence relation which rigorously defines continuous transformation from one path to another path with same end points.

**Definition A.3** (Homotopy). Let $f$ and $g$ be two paths in the graph $G$ which starts from a node $v_0$ and ends at a node $v_1$. Note that these paths can be considered as functions from the unit interval $[0, 1]$ to $G$, i.e. $f, g : [0, 1] \to G$ are paths such that $f(0) = g(0) = v_0$ and $f(1) = g(1) = v_1$. We say that two paths $f$ and $g$ are homotopic if there exists a continuous function $H : [0, 1] \times [0, 1] \to G$ such that

1. $H(t, 0) = f(t)$

2. $H(t, 1) = g(t)$

3. $H(0, x) = v_0$ and $H(1, x) = v_1$ for every $x \in [0, 1]$.

The homotopic relation on paths with the same end points in any space is an equivalence relation, see Chapter 1.1 of Hatcher (2002) for the proof of this nontrivial fact. Given a path $f : [0, 1] \to G$ over a graph $G$, we denote by $[f]$ the equivalence class of $f$ under the homotopy equivalence relation. Using this equivalence relation, we now construct the universal cover of a graph $G$.

**Definition A.4** (Universal Covering Space). Let $G$ be a graph. Fix a point $v \in V(G)$. The universal cover of $G$, denoted as $\tilde{G}$, is the space of homotopic classes of paths in $G$ starting at $v$:

$$\tilde{G} := \{[f] \mid f : [0, 1] \to G \text{ such that } f(0) = v\} \tag{11}$$

We end this subsection with the theorem that universal covers of connected graphs $G$ is a graph without any cycles, whose proof can be found in Chapter 1.3 of Hatcher (2002).

**Theorem A.5.** *Let $G$ be a connected graph. Then its universal cover $\tilde{G}$ is simply connected. In particular, it is a connected graph without any cycles.*

## A.2 GRAPH NEURAL NETWORKS

Given a graph $G := (V, E)$ with $n$ nodes, denote by $A \in \mathbb{R}^{n \times n}$ the adjacency matrix of $G$, $D \in \mathbb{R}^{n \times n}$ the diagonal matrix of node degrees of $G$, and $X \in \mathbb{R}^{n \times k}$ the matrix of concatenated $k$-dimensional node attributes of $G$. Denote by $\tilde{A} \in \mathbb{R}^{n \times n}$ the normalized adjacency matrix of $G$. (One may take, for instance, $\tilde{A} := D^{-\frac{1}{2}} A D^{-\frac{1}{2}}$.)

**Definition A.6.** Throughout the appendix, we use the notation $\{\{\cdot\}\}$ to denote a multiset of real vectors, i.e. we allow multiple instances of its elements.

*Remark* A.7. Let $\mathcal{M}_m^k$ be the collection of all multisets of $k$-dimensional vectors with $m$ elements, counting multiplicities. Suppose $F : \mathbb{R}^{k \times m} \to \mathbb{R}^l$ is a function which is invariant under the permutation action of the symmetric group with $m$ elements $S_m$. (The action corresponds to the permutation of rows of the $k \times m$ real matrix). Then the function $F$ induces the function over $\mathcal{M}_m^k$ defined as

$$\begin{aligned} \tilde{F} : \mathcal{M}_m^k &\to \mathbb{R}^l \\ \{\{v_1, v_2, \cdots, v_m\}\} &\mapsto F(v_1, v_2, \cdots, v_m) \end{aligned} \tag{12}$$

We recall the definition of graph neural networks proposed from Xu et al. (2018).

**Definition A.8.** We denote by $GNN^l$ the graph neural network comprised of composition of $l$ neighborhood aggregating layers.

1. Each $m$-th layer $H^{(m)}$ of the network constructs hidden node attributes of dimension $k_m$, denoted as $h_v^{(m)}$, using the following composition of functions:

$$\begin{cases} h_v^{(m)} := \text{COMBINE}^{(m)} \left( h_v^{(m-1)}, \text{AGGREGATE}_v^{(m)} \left( \left\{ \left\{ h_u^{(m-1)} \mid \begin{smallmatrix} u \in V(G), u \neq v \\ (u,v) \in E(G) \end{smallmatrix} \right\} \right\} \right) \right) \\ h_v^{(0)} := X_v \end{cases} \tag{13}$$

In the equation above, $X_v$ is the initial node attribute at $v$, $M_v^{(m)}$ is the collection of all multisets of $k_{m-1}$-dimensional real vectors with $\deg v$ elements counting multiplicities, the aggregation function

$$\text{AGGREGATE}_v^{(m)} : M_v^{(m)} \to \mathbb{R}^{k'_m} \tag{14}$$

is a set theoretic function of $k'_m$-dimensional real vectors defined over $M_v$, and the combination function

$$\text{COMBINE}^{(m)} : \mathbb{R}^{k_{m-1} + k'_m} \to \mathbb{R}^{k_m} \tag{15}$$

is a set theoretic function combining the attribute $h_v^{m-1}$ and the image of $\text{AGGREGATE}_v^{(m)}$.

2. Denote by $H^{(l)}$ the final layer of the network. Let $M^{(l)}$ be the collection of all multisets of $k_l$-dimensional vectors with $\#V(G)$ elements. Let

$$\text{READOUT} : M^{(l)} \to \mathbb{R}^K \tag{16}$$

be the graph readout function of $K$-dimensional real vectors defined over the multiset $M^{(l)}$. Then the $K$-dimensional vector representation of $G$, denoted as $h_G$, is given by

$$h_G := \text{READOUT}\left(\{\{h_v^{(l)} \mid v \in V(G)\}\}\right) \tag{17}$$

Observant readers may notice that graph neural network is a generalization of the color refinement algorithm, which is designed for distinguishing non-isomorphic pairs of graphs with identical discrete node labels Krebs & Verbitsky (2015).

### A.3 Hidden node attributes and universal covers

We now provide a rigorous formulation that graph neural networks compute vector representations of finite depth unfolding trees of a graph.

**Definition A.9.** Let $G := (V, E)$ be a graph. Denote by $G' := (V, E')$ the graph where every node has a self-loop. Denote by $G'' := (V, E'')$ the directed graph with a projection map $p : G'' \to G'$ such that each undirected edge of $G'$ from $v_1$ to $v_2$ corresponds to the following edges:

1. If $v_1 \neq v_2$, the undirected edge corresponds to two directed edges, one edge from $v_1$ to $v_2$, and the other from $v_2$ to $v_1$.

2. If $v_1 = v_2$, the edge corresponds to a single directed self-loop from $v_1$ to itself.

By construction, $k$-dimensional continuous node attributes over $G$, denoted as the function $f_G : V(G) \to \mathbb{R}^k$, clearly extends to continuous node attributes over $G''$, denoted as $f_{G''} : V(G'') \to \mathbb{R}^k$. One can also induce attributes over the set of nodes of the universal cover of $G$. We achieve this by pre-composing the function $f_G$ with the covering map $\pi_G : \tilde{G} \to G$.

**Definition A.10.** Let $f_G : V(G) \to \mathbb{R}^k$ be the function of $k$-dimensional node labels over the graph $G$. Let $\pi_G : \tilde{G} \to G$ be the universal covering map. Note that the covering map restricts to a function between set of nodes $\pi_G : V(\tilde{G}) \to V(G)$. The pullback of the node labels to the universal cover $\tilde{G}$ is the composition of functions $f_G \circ \pi_G : V(\tilde{G}) \to \mathbb{R}^k$. (See Figure 6 for an illustration of how pullback of node features are defined over the universal cover).

**Definition A.11.** Let $G := (V, E)$ be a directed graph. For each node $v \in V(G)$ and any positive number $k$, the depth-k neighborhood rooted at $v$, denoted as $U_v^k$, is a subgraph of $G$ whose set of nodes consists of the distinguished node $v$ itself and the nodes $w$ such that there exists at most $k$ consecutive directed edges from $v$ to $w$. The set of edges of $U_v^k$ are comprised of unions of all $k$ consecutive directed edges from $v$.

(For any undirected graph, the finite depth neighborhoods rooted at $v$ is defined in an analogous manner, where the set of edges is comprised of undirected edges among a pair of nodes of $G$, instead of directed edges).

Given a graph $G$, there exists an injective lift of set of nodes from $G$ to $\tilde{G}''$, defined as

$$i_G : V(G) \rightarrow_= V(G') \rightarrow_= V(G'') \to V(\tilde{G}'') \tag{18}$$

Likewise, there also exists an injective lift of set of nodes from $G$ to its universal cover $\tilde{G}$:

$$i_G^{un} : V(G) \to V(\tilde{G}). \tag{19}$$

Using a predetermined injective lift of nodes of $G$ to $\tilde{G}$ and $\tilde{G}''$, we may define finite depth unfolding trees at a node $v \in V(G)$ as follows.

**Definition A.12.** Let $G := (V, E)$ be a graph. Fix an injective lift $i_G : V(G) \to V(\tilde{G}'')$. The directed depth $k$ unfolding tree at a node $v \in V(G)$, denoted as $T_v^k$, is the depth-k neighborhood rooted at $i_G(v)$ as a subtree of the associated universal cover $\tilde{G}''$ of $G$. The undirected depth $k$ unfolding tree at a node $v \in V(G)$, denoted as $T_v^{k,un}$ is the depth-k neighborhood rooted at $i_G^{un}(v)$ as a subtree of the universal cover $\tilde{G}$ of $G$.

Throughout this manuscript, we will abbreviate the choice of injective lift from the base space $G$ to its covering space $\tilde{G}''$. This is thankfully because if $\tilde{G}''$ is an infinite graph, then the construction of universal covers imply that the directed depth k unfolding trees rooted at a fixed node $v \in V(G)$ obtained from any given injective lift $i_G : V(G) \to V(\tilde{G}'')$ are all isomorphic to each other. With abuse of notation, we will not specify the choice of injective lifts when defining depth k unfolding trees.

We note that the initial node attributes $X_v$ used for graph neural networks (Definition A.8) is given by

$$X_v = f_G(v) = (f_{G''} \circ \pi_{G''})(i_G(v)) \tag{20}$$

*Example* A.13. Let $G$ be an undirected graph without self-loops given as in Figure 3. The associated graphs $G'$ and $G''$ are given as in Figure 3. Let $f_G : V(G) \to \mathbb{R}^3$ be a function of 3-dimensional node attributes over $G$. In the exemplary figure, the given node attributes are coordinate-wise real vectors, each represented as discrete node labels, i.e. $(1,0,0) \mapsto A$, $(0,1,0) \mapsto B$, and $(0,0,1) \mapsto C$. Figures 1 and 5 demonstrates the directed depth 3 unfolding tree at a node, considered as a subspace of the universal cover $\tilde{G}''$ of $G''$.

*Remark* A.14. Given a node $v \in V(G)$ of an undirected graph $G := (V, E)$, the following three relations hold among directed depth $m$ unfolding subtrees of $T_v^{m+1}$.

1. For any two nodes $w_1, w_2$ adjacent to $v \in V(G)$ excluding $v$ itself, there exists a pair of disjoint subtrees $T_1, T_2$ of $T_v^{m+1} \subset \tilde{G}''$ which are isomorphic to directed depth $m$ unfolding trees rooted at $w_1$ and $w_2$, i.e.

$$T_{w_1}^m \cap T_{w_2}^m = \emptyset \tag{21}$$

2. The tree $T_v^{m+1}$ contains disjoint unions of directed depth $m$ unfolding trees rooted at all nodes adjacent to $v$ (including itself), i.e.

$$T_v^{m+1} \supset \bigsqcup_{\substack{w \in V(G) \\ (w,v) \in E(G)}} T_w^m \tag{22}$$

3. The set of nodes of the tree $T_v^{m+1}$ is the disjoint union of the singleton set $\{v\}$ and the set of nodes of depth $m$ unfolding trees at all nodes adjacent to $v$ (including itself), i.e.

$$V(T_v^{m+1}) = \{v\} \sqcup \bigsqcup_{\substack{w \in V(G) \\ (w,v) \in E(G)}} V(T_w^m) \tag{23}$$

Figure 5 illustrates how directed depth 2 unfolding subtrees of $T_v^3$ constructed from Figure 1 satisfy the three aforementioned relations. Note the depth 2 unfolding subtrees rooted at each node are marked in different colors.

The goal of this section is to identify hidden node attributes obtained from graph neural networks as a function from a multiset to real vector spaces. To do so, we define what is called the multiset of nested multisets associated to finite depth unfolding trees. Before doing so, we introduce some notational abbreviations.

**Definition A.15.** Given a multiset $S$ and number $d$, we use the abbreviation $\{\{S\}\}_d$ to denote the multiset whose elements are multisets of $d$ elements in $S$. Given a function of multisets $f : A \to B$, we denote by $\{\{f\}\} : A \to \{\{B\}\}$ the function obtained from representing each image of $f$ as a singleton multiset, i.e.

$$\{\{f\}\}(a) := \{\{f(a)\}\} \text{ for any } a \in A \tag{24}$$

Given two multisets $A, B$, the sum of two multisets, denoted as $A + B$, is a concatenation of $A$ and $B$, i.e. it is a multiset whose elements are in either $A$ or $B$, and whose element-wise multiplicity is the sum of multiplicities of elements of $A$ and $B$. One may use the usual summation notation $\sum$ to denote a sum of several multisets.

For example, an element of $\{\{\mathbb{R}^k\}\}_3$ is a multiset of 3 real vectors of dimension $k$. Given a real valued function $f(x) = 2x$, the function $\{\{f\}\}$ sends x to the singleton multiset $\{\{2x\}\}$. The sum of two multisets $\{\{2, 3, 5\}\}$ and $\{\{3, 6, 7\}\}$ is equal to $\{\{2, 3, 3, 5, 6, 7\}\}$.

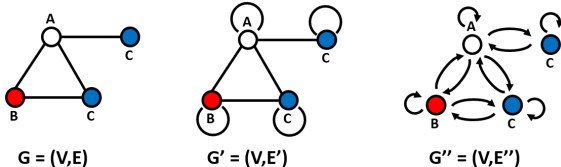

Figure 3: An exemplary graph $G$ with the induced graphs $G'$ and $G''$.

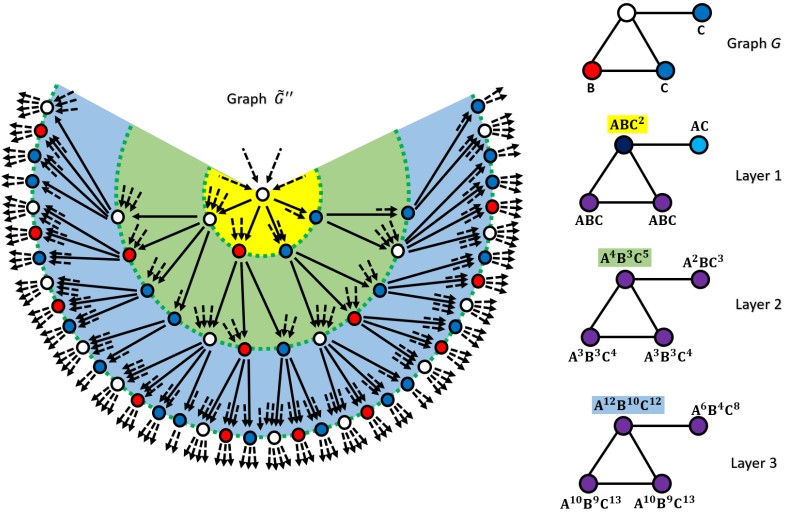

Figure 4: An illustration of Theorem A.19. Each respective node labels obtained from summing the attributes of the node $v$ itself and those of nodes adjacent to $v$ correspond to the sum of attributes of nodes of $\tilde{G}''$ in the respective colored regions.

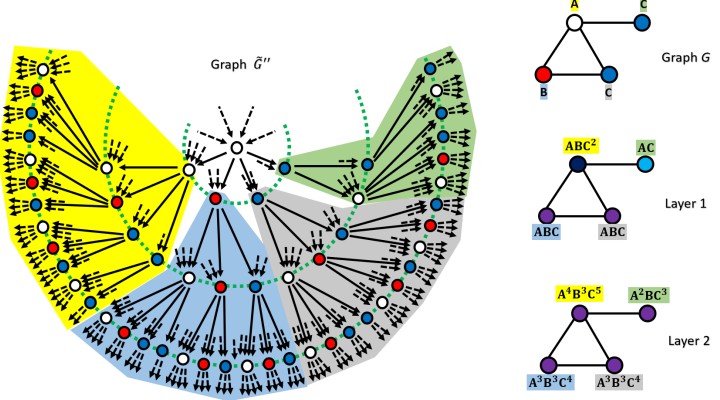

Figure 5: An illustration of three relations among depth 2 unfolding subtrees of $T_v^3$ from Remark A.14. Each respective node labels obtained from summing the attributes of the node $v$ itself and those of nodes adjacent to $v$ correspond to the sum of node labels of $T_v^m$ in the respective colored regions.

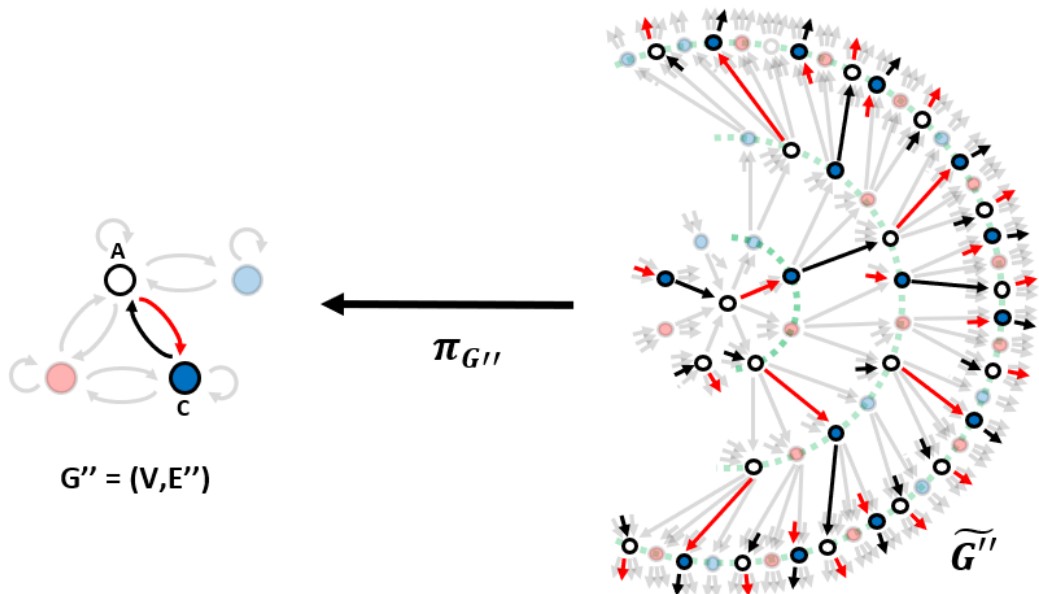

Figure 6: A visual demonstration of pullback of node labels from a given graph $G''$ to its universal cover $\tilde{G}''$, as defined from 3 and 4. The corresponding node attributes and edges are marked in identical colors.

**Definition A.16.** Let $G := (V, E)$ be a finite undirected graph without self-loops, $f_G : V(G) \to \mathbb{R}^k$ the function of $k$-dimensional node attributes over $G$, and $T_v^l$ the depth $l$ unfolding tree at $v \in V(G)$. We inductively define the multiset of nested multisets $\mathcal{T}_v^l$ associated to the depth $l$ unfolding tree $T_v^l$ as follows.

$$\mathcal{T}_v^l := \begin{cases} \mathbb{R}^k & \text{if } l = 0 \\ \sum_{u \in V(T_v^1) \setminus V(T_v^0)} \{\{\mathcal{T}_u^{l-1}\}\}_1 & \text{otherwise} \end{cases} \tag{25}$$

where $\sum$ is the multiset summation operator.

*Remark* A.17. We note that the multiset of nested multisets $\mathcal{T}_v^l$ is not identical to the multiset of of $k$-dimensional real vectors with $\# (V(T_v^l) \setminus V(T_v^{l-1}))$ elements. Nevertheless, recall from Remark A.14 that

$$V(T_v^l) \setminus V(T_v^{l-1}) = \bigsqcup_{u \in V(T_v^1) \setminus V(T_v^0)} V(T_u^{l-1}). \tag{26}$$

The above relation, along with the inductive construction of $\mathcal{T}_v^l$, induces an inductive construction of a morphism

$$p_{T_v^l} : \mathbb{R}^{k \times \#(V(T_v^l) \setminus V(T_v^{l-1}))} = \prod_{u \in V(T_v^1) \setminus V(T_v^0)} \mathbb{R}^{k \times \# V(T_u^{l-1})} \to \mathcal{T}_v^l \tag{27}$$

which sends a tuple of node attributes $((f_{G''} \circ \pi_{G''})(u))_{u \in V(T_v^l) \setminus V(T_v^{l-1})}$ supported over the set of nodes $V(T_v^l) \setminus V(T_v^{l-1})$ to the corresponding multiset respecting the subgraph structure specified by depth $l-1$ unfolding subtrees. Here, for each $u \in V(T_v^l) \setminus V(T_v^0)$, the vectors $L_u^{l-1}$ are elements of $\mathbb{R}^{k \times T_u^{l-1}}$, i.e. the concatenation of all $k$-dimensional node attributes supported over the depth $l-1$ unfolding tree rooted at $u$.

$$p_{T_v^l} \left( (L_u^{l-1})_{u \in V(T_v^1) \setminus V(T_v^0)} \right) = \begin{cases} f_G(v) & \text{if } l = 0 \\ \sum_{u \in V(T_v^1) \setminus V(T_v^0)} \left( \{\{p_{T_u^{l-1}}\}\}(L_u^{l-1}) \right) & \text{otherwise} \end{cases} \tag{28}$$

Note that $p_{T_v^0}$ is the identity function from $\mathbb{R}^k$ to itself. As will be demonstrated in the upcoming example, the functions $p_{T_v^l}$ are generalizations of the Weisfeiler-Lehman iteration scheme for updating node attributes.

*Example* A.18. Let $G$ be an undirected graph without self-loops given as in Figure 3. Let $f_G : V(G) \to \mathbb{R}^3$ be a function of 3-dimensional node attributes over $G$. Recall that the given node attributes are coordinate-wise real vectors, each represented as discrete node labels, i.e. $(1, 0, 0) \mapsto A$, $(0, 1, 0) \mapsto B$, and $(0, 0, 1) \mapsto C$.

Let $v \in V(G)$ be a node whose attribute is represented as $A$. Denote by $u_1 \in V(G)$ the node whose attribute is represented as $B$, $u_2 \in V(G)$ the degree 3 node whose attribute is represented at $C$, and $u_3$ the remaining node. Figure 4 illustrates the directed depth 3 unfolding tree $T_v^3$ rooted at the node $v$. We explain how the maps $p_{T_v^0}, p_{T_v^1}$, and $p_{T_v^2}$ are defined. The map $p_{T_v^0} : \mathbb{R}^3 \to \mathbb{R}^3$, by definition, is an identity function.

$$p_{T_v^0}(A) = A \tag{29}$$

The map $p_{T_v^1} : \mathbb{R}^{3 \times \#(V(T_v^1) \setminus V(T_v^0))} \to \mathcal{T}_v^1$ sends a tuple of 3-dimensional real vectors of length $\deg v + 1 = 4$ to the following multiset:

$$p_{T_v^1} : \mathbb{R}^{3 \times \#(V(T_v^1) \setminus V(T_v^0))} \to \mathcal{T}_v^1 = \sum_{u \in V(T_v^1) \setminus V(T_v^0)} \{\{\mathbb{R}^3\}\}_1 = \{\{\mathbb{R}^3\}\}_4$$

$$p_{T_v^1}((A, B, C, C)) = \{\{p_{T_v^0}(A)\}\} + \{\{p_{T_{u_1}^0}(B)\}\} + \{\{p_{T_{u_2}^0}(C)\}\} + \{\{p_{T_{u_3}^0}(C)\}\} \tag{30}$$

$$= \{\{A\}\} + \{\{B\}\} + \{\{C\}\} + \{\{C\}\} = \{\{A, B, C, C\}\}$$

The map $p_{T_v^2} : \mathbb{R}^{k \times \#(V(T_v^2) \setminus V(T_v^1))} \to \mathcal{T}_v^2$ sends a tuple of 3-dimensional real vectors of length $\#(V(T_v^2) \setminus V(T_v^1)) = 12$ to the following multiset of multisets:

$$p_{T_v^2} : \mathbb{R}^{3 \times \#(V(T_v^2) \setminus V(T_v^1))} \to \mathcal{T}_v^2 = \sum_{u \in V(T_v^1) \setminus V(T_v^0)} \{\{\mathcal{T}_u^1\}\}$$

$$p_{T_v^2}((A, B, C, C, A, B, C, A, B, C, A, C))$$
$$= \{\{p_{T_v^1}\}\}(A, B, C, C) + \{\{p_{T_{u_1}^1}\}\}(A, B, C) + \{\{p_{T_{u_2}^1}\}\}(A, B, C) + \{\{p_{T_{u_3}^1}\}\}(A, C) \tag{31}$$
$$= \{\{\{\{A, B, C, C\}\}\}\} + \{\{\{\{A, B, C\}\}\}\} + \{\{\{\{A, B, C\}\}\}\} + \{\{\{\{A, C\}\}\}\}$$
$$= \{\{\{\{A, B, C, C\}\}, \{\{A, B, C\}\}, \{\{A, B, C\}\}, \{\{A, C\}\}\}\}$$

Using these definitions, we are now able to rigorously formulate that graph neural networks capture local topological properties of $G$ by representing subtrees of the associated universal cover $\tilde{G}''$ as real vectors.

**Theorem A.19.** *Let $G := (V, E)$ be a graph. Denote by $f_G : V(G) \to \mathbb{R}^k$ the function of $k$-dimensional node labels over $G$. The node labels over $G$ can be extended to those over the graph $G''$, denoted as $f_{G''}$. Denote by $\pi_{G''} : \tilde{G}'' \to G''$ the universal covering map of $G''$. Let $T_v^k$ be the directed depth $k$ unfolding tree of the graph $G''$ at the node $v \in V(G'')$.*

*Let $GNN^l$ be a graph neural network comprised of $l$ layers. For each node $v \in V(G)$, let $\mathcal{T}_v^l$ be the multiset of nested multisets associated to the depth $l$ unfolding tree $T_v^l$, as constructed in Definition A.16. Let $p_{\mathcal{T}_v^l} : \mathbb{R}^{k \times \#(V(T_v^l) \setminus V(T_v^{l-1}))} \to \mathcal{T}_v^l$ be the morphism defined as in Remark A.17. Then there exists a set theoretic function $F_v^l : \mathcal{T}_v^l \to \mathbb{R}^{k_l}$ such that the node label at $v$ obtained from graph neural networks with $l$ layers given by*

$$h_v^{(l)} = F_v^l \left( p_{T_v^l} \left( ((f_{G''} \circ \pi_{G''})(u))_{u \in V(T_v^l) \setminus V(T_v^{l-1})} \right) \right) \tag{32}$$

*In other words, the updated node attributes are obtained from a set theoretic function supported over the set of nodes in $V(T_v^l) \setminus V(T_v^{l-1})$.*

*Proof.* We prove the theorem by induction. For any graph $G := (V, E)$ with a predetermined normalized adjacency matrix $\tilde{A}$, denote by $\tilde{a}_{u,v}$ the entry of the normalized adjacency matrix $\tilde{A}$ at a pair of nodes $(u, v)$.

Suppose $l = 1$. The node label at $v \in V(G)$ updated from a graph neural network with a single layer is obtained from taking a weighted sum of labels of nodes adjacent to $v$ and the node $v$ itself, followed by evaluating the newly obtained attributes using the activation function $\sigma_1$. By construction, the depth 0 unfolding tree of the graph $G''$ at $v$ is the node $v$ itself. The parent node of the depth 1

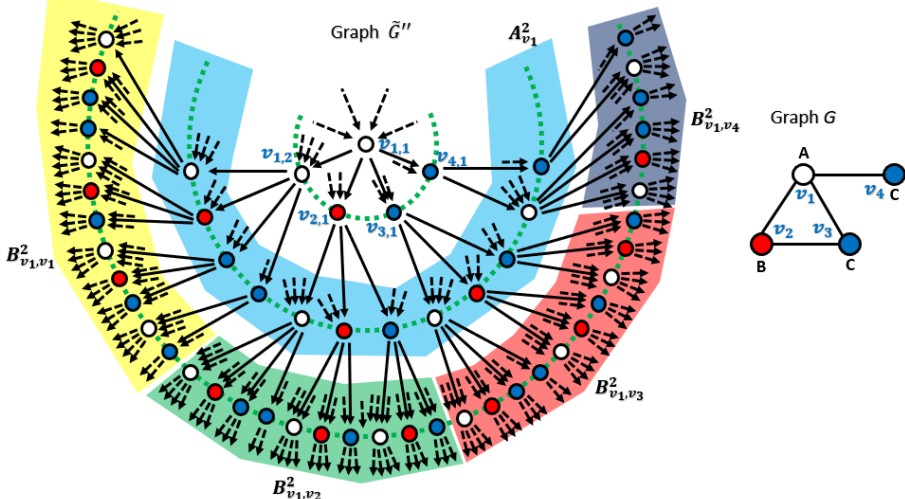

Figure 7: A visual representation of the set of nodes $A_u^m$ and $B_{v,u}^m$ from the exemplary graph $G$ from Figure 3. Observe that the lift of nodes $v_1 \in V(G)$ are notated as $v_{i,j}$, and the corresponding set of nodes $A_u^m$ and $B_{v,u}^m$ for $m = 2$ are shaded in distinct colors.

unfolding tree $T_v^1$ is the node $v$. The child nodes of $T_v^1$ are the nodes of $G$ adjacent to $v$, including $v$ itself. Hence there exists a bijection

$$\varphi_v^1 : (\{v\} \cup \{u \in V(G) \mid u \neq v, (u,v) \in E(G)\}) \to \left(V(T_v^1) \setminus V(T_v^0)\right) \tag{33}$$

which induces an equality of respective restrictions of node attributes $f_G$ and $f_{G''} \circ \pi_{G''}$. By definition and the existence of the bijection $\varphi_1$, there exists a function $F_v^1 : \mathcal{T}_v^1 \to \mathbb{R}^{k_1}$ such that

$$\begin{aligned}
h_v^{(1)} &= F_v^1 \left(\{\{(f_{G''} \circ \pi_{G''})(u) \mid u \in V(T_v^1) \setminus V(T_v^0)\}\}\right) \\
&= F_v^1 \left(p_{T_v^1} \left(((f_{G''} \circ \pi_{G''})(u))_{u \in V(T_v^1) \setminus V(T_v^0)}\right)\right)
\end{aligned} \tag{34}$$

Suppose the theorem holds for $l = m$. For any node $u$ adjacent to $v$ (i.e. $u \in \left(V(T_v^1) \setminus \{v\}\right)$), the three relations from Remark A.14 among directed finite depth unfolding trees establish a bijection of set of nodes $\varphi_u^m : A_u^m \to B_{v,u}^m$, where the sets $A_u^m$ and $B_{v,u}^m$ are given as

$$\begin{aligned}
A_u^m &= V(T_u^m) \setminus V(T_u^{m-1}) \\
B_{v,u}^m &= V(T_v^{m+1}) \setminus \left(V(T_v^m) \bigcup \left(\bigsqcup_{w \in (A_u^1 \setminus \{u\})} A_w^m\right)\right)
\end{aligned} \tag{35}$$

The bijection $\varphi_u^m$ induces an equality between respective restrictions of pullback of node attributes $f_{G''} \circ \pi_{G''}$. The collection of bijections $\{\varphi_u^m\}_{u \in A_v^1}$ extends to bijections

$$\begin{aligned}
\varphi^{m+1} &: \bigsqcup_{u \in A_v^1} B_{v,u}^m \to A_v^{m+1} \\
\tilde{\varphi}^{m+1} &: \bigsqcup_{u \in A_v^1 \setminus \{v\}} B_{v,u}^m \to A_v^{m+1} \setminus A_v^m
\end{aligned} \tag{36}$$

both of which induce equalities between respective restrictions of $f_{G''} \circ \pi_{G''}$. We refer to Figure 7 for an illustration of the set of nodes $A_u^m$ and $B_{v,u}^m$ obtained from the exemplary graph drawn in Figure 3.

Denote by $\tilde{\mathcal{T}}_v^{m+1}$ the following multiset of nested multisets:

$$\tilde{\mathcal{T}}_v^{m+1} := \sum_{\substack{u \in V(T_v^1) \\ \pi_{G''}(u) \neq v}} \{\{\mathcal{T}_u^m\}\}_1 \tag{37}$$

In other words, $\tilde{\mathcal{T}}_v^{m+1}$ is a multiset summation of nested multisets associated to depth $m$ unfolding trees $T_u^m$ rooted at all nodes $u$ which do not map to the given node $v$ under the universal covering map $\pi_{G''} : \tilde{G}'' \to G''$. Denote by $\tilde{p}_{T_v^{m+1}} : \mathbb{R}^{k \times \#(A_v^{m+1} \setminus A_v^m)} \to \tilde{\mathcal{T}}_v^{m+1}$ the function given by:

$$\tilde{p}_{T_v^{m+1}} \left( (L_u^m)_{u \in A_v^{m+1} \setminus A_v^m} \right) := \sum_{u \in A_v^{m+1} \setminus A_v^m} \left( \{\{ p_{T_u^m} \}\}(L_u^m) \right). \tag{38}$$

Here, for each node $u$, the vectors $L_u^m$ are elements of $\mathbb{R}^{k \times T_u^m}$, i.e. the concatenation of all $k$-dimensional node attributes supported over the depth $m$ unfolding tree rooted at $u$. Note that as a multi-set function,

$$p_{T_v^{m+1}} \left( (L_u^m)_{u \in V(T_v^1) \setminus V(T_v^0)} \right) = p_{T_v^m}(L_v^m) + \tilde{p}_{T_v^{m+1}} \left( (L_u^m)_{u \in A_v^{m+1} \setminus A_v^m} \right) \tag{39}$$

For some positive number $k'_{m+1}$ there exists a function $\tilde{F}_v^{m+1} : \tilde{\mathcal{T}}_v^{m+1} \to \mathbb{R}^{k'_{m+1}}$ such that

$$\text{AGGREGATE}_v^{(m+1)} \left( \left\{ \left\{ h_u^{(m)} \mid {}^{u \in V(G), u \neq v}_{(u,v) \in E(G)} \right\} \right\} \right)$$
$$= \tilde{F}_v^{m+1} \left( \tilde{p}_{T_v^{m+1}} \left( ((f_{G''} \circ \pi_{G''})(u))_{u \in A_v^{m+1} \setminus A_v^m} \right) \right). \tag{40}$$

To see this, we observe that

$$\text{AGGREGATE}_v^{(m+1)} \left( \left\{ \left\{ h_u^{(m)} \mid {}^{u \in V(G), u \neq v}_{(u,v) \in E(G)} \right\} \right\} \right)$$
$$= \text{AGGREGATE}_v^{(m+1)} \left( \left\{ \left\{ F_u^m \left( p_{T_u^m} \left( ((f_{G''} \circ \pi_{G''})(w))_{w \in B_{v,u}^m} \right) \right) \mid {}^{u \in V(G), u \neq v}_{(u,v) \in E(G)} \right\} \right\} \right)$$
$$= \text{AGGREGATE}_v^{(m+1)} \left( \left\{ \left\{ F_u^m \left( p_{T_u^m} \left( ((f_{G''} \circ \pi_{G''})(w))_{w \in B_{v,u}^m} \right) \right) \mid u \in A_v^1 \setminus \{v\} \right\} \right\} \right) \tag{41}$$
$$= \tilde{F}_v^{m+1} \left( \tilde{p}_{T_v^{m+1}} \left( ((f_{G''} \circ \pi_{G''})(u))_{u \in A_v^{m+1} \setminus A_v^m} \right) \right)$$

The last equation follows from the following observation. The domain of the aggregation function is supported over the set of nodes $\bigsqcup_{u \in A_v^1 \setminus \{v\}} B_{v,u}^m$. Applying the bijection $\tilde{\varphi}^{m+1}$ implies that the aggregation function is defined over the real vector space supported over the set of nodes $A_v^{m+1} \setminus A_v^m$.

By (39), we hence obtain that there exists a function $F_v^{m+1} : \mathcal{T}_v^{m+1} \to \mathbb{R}^{k_{m+1}}$ such that

$$h_v^{(m+1)} = \text{COMBINE}^{(m+1)} \left( h_v^{(m)}, \ \text{AGGREGATE}_v^{(m)} \left( \left\{ \left\{ h_u^{(m)} \mid {}^{u \in V(G), \ u \neq v}_{(u,v) \in E(G)} \right\} \right\} \right) \right)$$
$$= \text{COMBINE}^{(m+1)} \left( h_v^{(m)}, \tilde{F}_v^{m+1} \left( \tilde{p}_{T_v^{m+1}} \left( ((f_{G''} \circ \pi_{G''})(u))_{u \in A_v^{m+1} \setminus A_v^m} \right) \right) \right)$$
$$= \text{COMBINE}^{(m+1)} \left( F_v^m \left( p_{T_v^m} \left( ((f_{G''} \circ \pi_{G''})(u))_{u \in A_v^m} \right) \right), \tilde{F}_v^{m+1} \left( \tilde{p}_{T_v^{m+1}} \left( ((f_{G''} \circ \pi_{G''})(u))_{u \in A_v^{m+1} \setminus A_v^m} \right) \right) \right)$$
$$= F_v^{m+1} \left( p_{T_v^{m+1}} \left( ((f_{G''} \circ \pi_{G''})(u))_{u \in A_v^{m+1}} \right) \right). \tag{42}$$

$\square$

*Example* A.20. We revisit the graph $G$ as shown in Figure 3. Consider a graph neural network with $m$ layers such that for all layers the aggregation function $\text{AGGREGATE}_v$ and the combination function COMBINE correspond to summation of respective node attributes. One may consider the resulting GNN as a simplified generalization of the Weisfeiler-Lehman isomorphism test for continuous node attributes. Concatenations of adjacent discrete node labels correspond to summations of adjacent node attributes, whereas substitutions of newly obtained node labels are skipped. Then there exists a correspondence between the node attributes updated from the graph neural network with $m$ layers and the attributes over the set of nodes in $V(T_v^m) \setminus V(T_v^{m-1})$. For example, as indicated in Figure 4 and 5, the updated node attributes can be obtained by counting the number of occurrences of attributes in the respective colored region.

## A.4 PROOF OF THEOREM 3.3

Xu et al. shows that graph neural networks with injective node feature aggregating functions and injective graph-level readout functions are as powerful as Weisfeiler-Lehman isomorphism tests in distinguishing non-isomorphic classes of graphs Xu et al. (2018). Using the theory of covering spaces, we prove a refinement of the isomorphism-invariant properties of Weisfeiler-Lehman tests over graphs with identical node labels Krebs & Verbitsky (2015) for graph neural networks.

Under certain conditions, the vector representations of graphs obtained from graph neural network can distinguish isomorphism classes of universal covers of graphs as well as non-equivalent pullback of node labels.

**Theorem A.21** (Theorem 3.3). *Let $\mathcal{G}$ be a collection of finite connected graphs such that the least upper bound of their diameters is equal to $d$. Suppose a graph neural network $GNN^l$ with $l$ layers satisfies the following three constraints:*

- *For every $m$ such that $1 \leq m \leq l$ and for each $v \in V(G)$, the functions $AGGREGATE_v^{(m)}$ and $COMBINE^{(m)}$ are injective.*

- *The graph read-out function READOUT is injective.*

- *$l \geq 2d$.*

*Then $GNN^l$ maps any two connected graphs $G, H \in \mathcal{G}$ with the same number of nodes to identical vector representations if and only if there exists an isomorphism of universal covers $\varphi : \tilde{G}'' \to \tilde{H}''$ that induces an equality of the pullback of node labels, i.e.*

$$h_G = h_H \iff \exists \, isomorphism \, \varphi : \tilde{G}'' \to \tilde{H}'' s.t. \, f_{G''} \circ \pi_{G''} = f_{H''} \circ \pi_{H''} \circ \varphi \qquad (43)$$

The theorem generalizes the result of Krebs and Verbitsky Krebs & Verbitsky (2015) for pairs of graphs with fixed constant node labels. Before we prove Theorem 3.3, we recall the definition of graph diameters and note the following lemma .

**Definition A.22** (Graph Diameter). Let $G := (V, E)$ be a finite undirected graph. Given a pair of nodes $v, w \in V(G)$, let $P_{v,w}$ be the set of sequences of edges $\mathfrak{p} = (e_i)_{i=1}^p$ such that one can move from node $v$ to node $w$ by traveling along the sequence of edges $(e_i)_{i=1}^p$. The length of the path $\mathfrak{p}$ is the size of the sequence, i.e. $l(\mathfrak{p}) = l((e_i)_{i=1}^p) = p$. Then the graph diameter $d_G$ of $G$ is the maximum of the shortest length of paths required to move between any two nodes. In other words,

$$d_G := \max_{(v,w) \in V(G)} \left( \min_{\mathfrak{p} \in P_{v,w}} l(\mathfrak{p}) \right) \qquad (44)$$

**Lemma A.23.** *Let $G_1$ and $G_2$ be two undirected graphs with finite number of nodes. Let $f_{G_1} : V(G_1) \to \mathbb{R}^k$ and $f_{G_2} : V(G_2) \to \mathbb{R}^k$ be node attribute functions over $G_1$ and $G_2$. Denote by $\pi_{G_i} : \tilde{G}_i \to G_i$ and $\pi_{G_i''} : \tilde{G}_i'' \to G_i''$ the universal covering maps for $i = 1, 2$. Then the isomorphism $h : \tilde{G}_1 \to \tilde{G}_2$ induces an isomorphism $h'' : \tilde{G}_1'' \to \tilde{G}_2''$, and vice versa. If such isomorphisms exist, then $f_{G_1} \circ \pi_{G_1} = f_{G_2} \circ \pi_{G_2} \circ h$ if and only if $f_{G_1''} \circ \pi_{G_1''} = f_{G_2''} \circ \pi_{G_2''} \circ h''$.*

*Proof.* Let $v \in V(G)$ be a node of an undirected graph $G$. We observe that a collection of finite depth unfolding trees rooted at all nodes of a universal cover $\tilde{G}$ (or $\tilde{G}''$) defines an open cover. It hence suffices to show that for any $l$, two undirected depth $l$ unfolding trees $T_v^{l,un}$ are isomorphic if and only if two directed depth $l$ unfolding trees $T_v^l$ are isomorphic. Note that by the construction of finite depth unfolding trees, the result on equality of node attributes follows immediately.

As constructed from Definition A.12, we denote by $T_v^{1,un} \subset \tilde{G}$ the undirected depth 1 unfolding tree rooted at $v$, and $T_v^1 \subset \tilde{G}''$ the directed depth 1 unfolding tree rooted at $v$. Note that the directed tree $T_v^1$ can be constructed from the undirected tree $T_v^{1,un}$ by substituting all undirected edges with directed edges from $v$ to its neighboring nodes, adding a new copy of the node $v$ itself, and adding a new directed edge from $v$ to its copy. This demonstrates that the statement holds for $l = 1$.

Suppose the isomorphism invariance of unfolding trees holds for $l = k$. As before, denote by $T_v^{k+1,un} \subset \tilde{G}$ the undirected depth $k + 1$ unfolding tree rooted at $v$, and $T_v^{k+1} \subset \tilde{G}''$ the directed

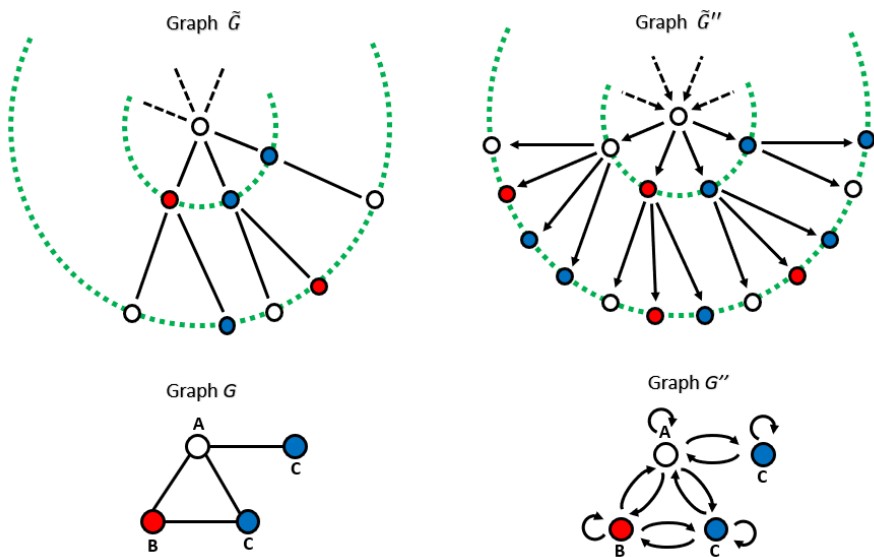

Figure 8: A comparison between the universal cover $\tilde{G}$ and the induced universal cover $\tilde{G}''$ of a graph $G$ constructed in Figure 3.

depth $k+1$ unfolding tree rooted at $v$. The directed tree $T_v^{k+1}$ can be constructed from the undirected tree $T_v^{k+1,un}$ using the following procedure. For every node $w \in V(T_v^{1,un}) \setminus \{v\}$, substitute the undirected tree $T_w^{k,un}$ rooted at $w$ with the directed tree $T_w^k$. We then add a disjoint copy of the directed tree $T_v^k$, and add a new directed edge from the root of the tree $T_v^{k+1}$ to the root of the newly added tree $T_v^k$ (This corresponds to adding a new directed edge from $v$ to its copy). The procedure guarantees to map isomorphic undirected unfolding trees to isomorphic directed unfolding trees, and vice versa. A visual illustration which compares the undirected and the directed rooted unfolding trees can be found in Figure 8.

$\square$

We recall the following two lemmas on isomorphism classes of finite depth unfolding trees. Both lemmas are reformulations of Lemma 2.5 and Lemma 2.7 of Krebs & Verbitsky (2015), which proves the statements for undirected finite depth unfolding rooted trees.

**Lemma A.24.** *Let $v, w \in V(G)$ be any pair of nodes such that for any $l \geq 1$, the following conditions hold.*

1. *The two directed unfolding trees $T_v^{l-1}$ and $T_w^{l-1}$ are isomorphic.*

2. *There exists a bijection of nodes $g_l : V(T_v^1) \to V(T_w^1)$ such that for all $u \in V(T_v^1)$, the two directed unfolding trees $T_u^l$ and $T_{g_l(u)}^l$ are isomorphic.*

*Then the directed unfolding trees $T_v^{l+1}$ and $T_w^{l+1}$ are isomorphic.*

**Lemma A.25.** *Let $G, H$ be two connected graphs with at most $n$ nodes. Let $d$ be the maximum of the diameters of two graphs $G$ and $H$, i.e. $d := \max(d_G, d_H)$. Then for any pair of nodes $v \in V(G)$ and $w \in V(H)$, the directed unfolding trees $T_v^l$ and $T_w^l$ are isomorphic for any $l \geq 2d$ if and only if $T_v^{2d-1}$ and $T_w^{2d-1}$ are isomorphic.*

*Proof.* Note that any depth-$d$ unfolding tree rooted at a node $v \in V(G)$ or $w \in V(H)$, as a subspace of the universal covers $\tilde{G}''$ and $\tilde{H}''$, contains all pre-images of the set of nodes $V(G)$ and $V(H)$. This implies that any depth-$2d - 1$ unfolding trees rooted at any node $v \in V(G)$ or $w \in V(H)$ contains all possible rooted depth-$d$ unfolding trees. It remains to invoke the ideas of the proof of Lemma 2.7 of Krebs & Verbitsky (2015) to further reduce the lower bound of $l$ from $2n$ to $2d$. $\square$

Using the aforementioned lemmas, we are able to prove that the hidden node attributes obtained from GNNs with $l$ layers whose combination and neighborhood aggregation functions are injective indicate the isomorphism classes of depth $l$ unfolding rooted trees.

**Lemma A.26.** *Let $G$ and $H$ be connected graphs with $n$ nodes. Denote by $f_G : V(G) \to \mathbb{R}^k$ and $f_H : V(H) \to \mathbb{R}^k$ the $k$-dimensional node attributes of $G$ and $H$. Denote by $\pi_{G''} : \tilde{G}'' \to G''$ and $\pi_{H''} : \tilde{H}'' \to H''$ the universal covering maps of the induced directed graphs $G''$ and $H''$.*

*Suppose a graph neural network $GNN^l$ with $l$ layers satisfies the condition that for every $m$ such that $1 \le m \le l$ and for each node $\hat{v}$ of any finite graph $\hat{G}$, the functions $AGGREGATE_{\hat{v}}^{(m)}$ and $COMBINE^{(m)}$ are injective.*

*For each $v \in V(G)$, pick a bijection of set of nodes $\phi_v : V(G) \to V(H)$ which induces an equality (not an isomorphism) of depth-1 unfolding tree at $v$ and $\phi_v(v)$, i.e. $\phi_v(T_v^1) = \phi_v(T_{\phi_v(v)}^1)$. Then for every $1 \le m \le l$, the bijection $\phi_v$ induces an equality of hidden node attributes $h_v^{(m)}$ and $h_{\phi_v(v)}^{(m)}$ obtained from $GNN^l$ for every node $v \in V(G)$ if and only if the bijection $\phi_v$ induces isomorphisms $\varphi_{v,m} : T_v^m \to T_{\phi_v(v)}^m$ that imply equality of node attributes over the trees $T_v^m$ and $T_{\phi_v(v)}^m$. In other words:*

$$\begin{array}{c} \exists\ \varphi_{v,m}:T_v^m \xrightarrow{\cong} T_{\phi_v(v)}^m\ \text{such that} \\ (f_{G''}\circ\pi_{G''})|_{T_v^m}=(f_{H''}\circ\pi_{H''})|_{T_{\phi_v(v)}^m}\circ\varphi_{v,m} \end{array} \iff h_v^{(m)} = h_{\phi_v(v)}^{(m)} \quad \forall 1 \le m \le l. \tag{45}$$

Note that the bijections $\{\phi_v\}_{v \in V(G)}$ may not necessarily be a singleton set. We refer to Figure 9 for an example of a pair of graphs $G$ and $H$ where one needs to choose distinct bijections $\phi_v$ for each node $v$.

*Proof.* This lemma is a generalization of Lemma 2.6 of Krebs & Verbitsky (2015). We prove the statement of the lemma by induction on $l$. The base case for $l = 0$ is trivial, as the statement of the lemma simplifies to comparing the attributes of a given pair of nodes.

Suppose the equivalence relation holds up to $l = l_0$. For each $v \in V(G)$, let $\phi_v : V(G) \to V(H)$ be a bijection of set of nodes which induces an isomorphism of depth-1 unfolding trees rooted at $v$ and $\phi_v(v)$. Suppose the hidden node attributes obtained from $GNN$ with at most $l_0 + 1$ layers are identical for every node, i.e. $h_v^{(m)} = h_{\phi_v(v)}^{(m)}$ for $1 \le m \le l_0 + 1$. By the induction hypothesis, for any $1 \le m \le l_0$, the following equivalence relation holds for each pair of nodes $\{(v, \phi_v(v))\}_{v \in V(G)}$:

$$\begin{array}{c} \exists\ \varphi_{v,m}:T_v^m \xrightarrow{\cong} T_{\phi_v(v)}^m\ \text{such that} \\ (f_{G''}\circ\pi_{G''})|_{T_v^m}=(f_{H''}\circ\pi_{H''})|_{T_{\phi_v(v)}^m}\circ\varphi_{v,m} \end{array} \iff h_v^{(m)} = h_{\phi_v(v)}^{(m)} \tag{46}$$

Observe that for each $1 \le i \le l_0$, the isomorphism $\varphi_{v,i} : T_v^i \to T_{\phi_v(v)}^i$ induces the following equivalence relations for all pairs of nodes $\{(u, \varphi_{v,i}(u))\}_{u \in V(T_v^1) \setminus V(T_v^0)}$ and for any $1 \le m \le l_0$:

$$\begin{array}{c} \exists\ \varphi_{u,m}:T_u^m \xrightarrow{\cong} T_{\varphi_{v,i}(u)}^m\ \text{such that} \\ (f_{G''}\circ\pi_{G''})|_{T_u^m}=(f_{H''}\circ\pi_{H''})|_{T_{\varphi_{v,i}(u)}^m}\circ\varphi_{u,m} \end{array} \iff h_{\pi_{G''}(u)}^{(m)} = h_{\pi_{H''}(\varphi_{v,i}(u))}^{(m)} \tag{47}$$

Note that $\varphi_{v,i}(u) = \phi_v(u)$ because $\phi_v$ induces an isomorphism of depth-1 unfolding trees at $v$ and $\phi_v(v)$.

Consider the open cover of $T_v^{l_0+1}$ by the directed trees $\{T_u^{l_0}\}_{u \in V(T_v^1)}$. The intersection of any two trees satisfy

$$T_u^{l_0} \cap T_{u'}^{l_0} = \begin{cases} T_{u'}^{l_0-1} & \text{if } u \in V(T_v^0) \text{ and } u' \in V(T_v^1) \setminus V(T_v^0) \\ \emptyset & \text{otherwise} \end{cases} \tag{48}$$

By Lemma A.24, the collection of isomorphisms

$$\{\varphi_{u,l_0}\}_{u \in V(T_v^1)} \cup \{\varphi_{u,l_0-1}\}_{u \in V(T_v^1) \setminus V(T_v^0)} \tag{49}$$

induces the isomorphism $\varphi_{v,l_0+1} : T_v^{l_0+1} \to T_{\phi_v(v)}^{l_0+1}$.

Recall that the hidden node attributes $h_v^{(l_0+1)}$ and $h_{\phi_v(v)}^{(l_0+1)}$ are identical. Then the following collections of multiset of node attributes are identical.

$$p_{T_v^{l_0+1}}\left(\left((f_{G''} \circ \pi_{G''})(v)\right)_{v \in V(T_v^{l_0+1}) \setminus V(T_v^{l_0})}\right) = p_{T_{\phi_v(v)}^{l_0+1}}\left(\left((f_{H''} \circ \pi_{H''})(u)\right)_{u \in V(T_{\phi_v(v)}^{l_0+1}) \setminus V(T_{\phi_v(v)}^{l_0})}\right) \tag{50}$$

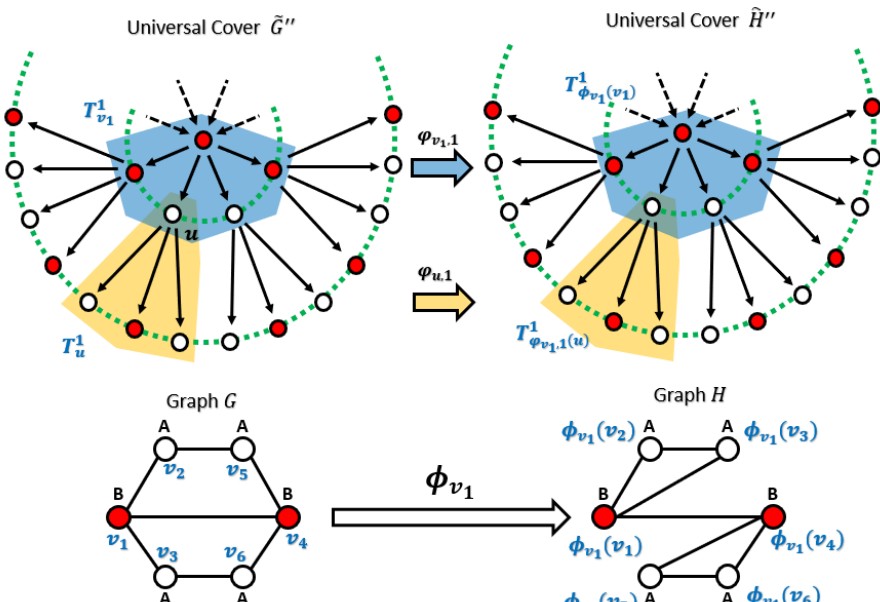

Figure 9: A visual demonstration of the gluing procedure from the proof of Theorem A.26. One can construct an isomorphism between a pair of depth $l+1$ unfolding trees rooted at $v$ and $\phi_v(v)$ by identifying the isomorphism classes of depth $l$ unfolding trees rooted at $u$ and $\phi_v(u)$ for all $u \in V(T_v^1)$ and gluing the trees in accordance to their intersections, which are depth $l-1$ unfolding trees (48). Observe that the bijection of the set of nodes $\phi_{v_1}$ does not induce an equality (not an isomorphism) of depth-1 trees at $v_i$ and $\phi_{v_1}(v_i)$ for $i=2,3,5,6$. For example, the set of nodes of the undirected depth-1 unfolding tree at $v_2$ is given by $\{v_1, v_2, v_5\}$, whereas the set of nodes of the undirected depth-1 unfolding tree at $\phi_{v_1}(v_2)$ is the set $\{\phi_{v_1}(v_1), \phi_{v_1}(v_2), \phi_{v_1}(v_3)\}$. For such nodes, different choices of the bijections of the set of nodes $\phi_{v_i}$ are required.

The above equation follows from the condition that the functions $\text{AGGREGATE}_v^{(m)}$ and $\text{COMBINE}_v^{(m)}$ are injective for all $v \in V(G)$ and $1 \le m \le l_0 + 1$, which implies that the function over the multiset of labels $F_v^{l_0+1}$ as constructed from Theorem A.19 is an injective function. The collection of isomorphisms from (49) further indicates that for any $u \in V(T_v^1) \setminus V(T_v^0)$, the following equality of node attributes over $T_v^{l_0}$, $T_u^{l_0}$, and their intersection $T_u^{l_0-1}$,

$$\left(f_{G''} \circ \pi_{G''}\right)\big|_{T_v^{l_0}} = \left(f_{H''} \circ \pi_{H''}\right)\big|_{T_{\phi_v(v)}^{l_0}} \circ \varphi_{v,l_0}$$

$$\left(f_{G''} \circ \pi_{G''}\right)\big|_{T_u^{l_0}} = \left(f_{H''} \circ \pi_{H''}\right)\big|_{T_{\phi_v(u)}^{l_0}} \circ \varphi_{v,l_0}$$

$$\left(f_{G''} \circ \pi_{G''}\right)\big|_{T_v^{l_0} \cap T_u^{l_0}} = \left(f_{G''} \circ \pi_G''\right)\big|_{T_u^{l_0-1}} \tag{51}$$

$$= \left(f_{H''} \circ \pi_{H''}\right)\big|_{T_{\phi_v(u)}^{l_0-1}} \circ \varphi_{u,l_0-1} = \left(f_{H''} \circ \pi_{H''}\right)\big|_{T_{\phi_v(v)}^{l_0} \cap T_{\phi_v(u)}^{l_0}} \circ \varphi_{u,l_0-1}$$

extends to the equality of node attributes over $T_v^{l_0} \cup T_u^{l_0}$:

$$\left(f_{G''} \circ \pi_{G''}\right)\big|_{T_v^{l_0} \cup T_u^{l_0}} = \left(f_{H''} \circ \pi_{H''}\right)\big|_{T_{\phi_v(v)}^{l_0} \cup T_{\phi_v(u)}^{l_0}} \circ \varphi_{v,l_0+1}\big|_{T_v^{l_0} \cup T_u^{l_0}}. \tag{52}$$

Iterating the gluing procedure for all depth $l_0$ trees $\{T_u^{l_0}\}_{u \in V(T_v^1) \setminus V(T_v^0)}$ results in the desired equality of node attributes over $T_v^{l_0+1}$.

Now suppose that there exists an equality of node attributes between $T_v^{l_0+1}$ and $T_{\phi_v(v)}^{l_0+1}$ induced from the unfolding tree isomorphism $\varphi_{v,l_0+1} : T_v^{l_0+1} \to T_{\phi_v(v)}^{l_0+1}$. Theorem A.19 implies that there exists a set theoretic function $F_v^{l_0+1} : \mathcal{T}_v^{l_0+1} \to \mathbb{R}^{k_{l_0+1}}$ such that the node label at $v$ obtained from graph neural networks with $l$ layers is given by

$$h_v^{(l_0+1)} = F_v^{l_0+1}\left(p_{T_v^{l_0+1}}\left(\left((f_{G''} \circ \pi'')(u)\right)_{u \in V(T_v^{l_0+1}) \setminus V(T_v^{l_0})}\right)\right). \tag{53}$$

The conditions that AGGREGATE$_v^{(l_0+1)}$ and COMBINE$_v^{(l_0+1)}$ are injective imply that $F_v^{l_0+1}$ is an injective function. Therefore, the equality of hidden node attributes $h_v^{(l_0+1)} = h_{\phi_v(v)}^{(l_0+1)}$ follows immediately from the fact that the equality of node attributes between depth $l_0 + 1$ unfolding trees ensures the equality of collection of multiset of node attributes over $V(T_v^{l_0+1}) \setminus V(T_v^{l_0})$. □

Using Lemma A.25 and A.26, we now prove Theorem 3.3.

*Theorem A.21 (Theorem 3.3).* For each $v \in V(G)$, let $\phi_v : V(G) \to V(H)$ be a bijection of set of nodes of $G$ and $H$ which induces an equality of depth-1 unfolding trees at $v$ and $\phi_v(v)$. Lemma A.26 implies that the following equivalence relation holds for every $v \in V(G)$:

$$\begin{array}{c} \exists\, \varphi_{v,m}:T_v^m \xrightarrow{\cong} T_{\phi_v(v)}^m \text{ such that} \\ (f_{G''} \circ \pi_{G''})|_{T_v^m} = (f_{H''} \circ \pi_{H''})|_{T_{\phi_v(v)}^m} \circ \varphi_{v,m} \end{array} \iff h_v^{(m)} = h_{\phi_v(v)}^{(m)} \quad \forall v \in V(G) \text{ and } 1 \le m \le 2d. \quad (54)$$

Lemma A.25 implies that for any positive number $l \ge 2d$, the following equivalence relation holds:

$$\begin{array}{c} \exists\, \varphi_{v,l}:T_v^l \xrightarrow{\cong} T_{\phi_v(v)}^l \text{ such that} \\ (f_{G''} \circ \pi_{G''})|_{T_v^l} = (f_{H''} \circ \pi_{H''})|_{T_{\phi_v(v)}^l} \circ \varphi_{v,l} \end{array} \iff h_v^{(m)} = h_{\phi_v(v)}^{(m)} \quad \forall v \in V(G) \text{ and } 1 \le m \le 2d. \quad (55)$$

Consider the open cover $\{T_w^l\}_{w \in V(\tilde{G}'')}$ consisting of directed depth $l$ unfolding trees rooted at every node of $\tilde{G}''$. Note that for any $w \in V(\tilde{G}'')$, there exists a node $v \in V(G)$ such that $T_w^l \cong T_v^l$. For any two nodes $w_1, w_2 \in V(\tilde{G})$, there exist injective lifts $i_G(w_1)$ and $i'_G(w_2)$ whose corresponding depth $l$ unfolding trees rooted at $w_i$'s satisfy the following equation:

$$T_{w_1}^l \cap T_{w_2}^l = \begin{cases} T_{w_2}^m & \text{if } \exists\ m \text{ consecutive directed edges from } i_G(w_1) \text{ to } i'_G(w_2) \\ T_{w_1}^m & \text{if } \exists\ m \text{ consecutive directed edges from } i'_G(w_2) \text{ to } i_G(w_1) \\ 0 & \text{otherwise} \end{cases} \quad (56)$$

Hence there exists an isomorphism $\varphi : \tilde{G}'' \to \tilde{H}''$ if and only if for some $l \ge 2d$, there exists a bijection $\phi : V(G) \to V(H)$ which induces an isomorphism of directed depth $l$ unfolding trees $\varphi_{v,l} : T_v^l \cong T_{\phi(v)}^l$ for every $v \in V(G)$. Furthermore, the induced node attributes $f_{G''} \circ \pi_{G''}$ and $f_{H''} \circ \pi_{H''}$ are identical if and only if for some $l \ge 2d$, there exists a bijection $\phi_v : V(G) \to V(H)$ which induces an equality of node attributes

$$(f_{G''} \circ \pi_{G''})|_{T_v^l} = (f_{H''} \circ \pi_{H''})|_{T_{\phi_v(v)}^l} \circ \varphi_{v,l} \quad (57)$$

imposed on directed depth $l$ unfolding trees $T_v^l \cong T_{\phi_v(v)}^l$ for each $v \in V(G)$. Therefore, we obtain that

$$\begin{array}{c} \exists\, \varphi:\tilde{G}'' \xrightarrow{\cong} \tilde{H}'' \text{ such that} \\ f_{G''} \circ \pi_{G''} = f_{H''} \circ \pi_{H''} \circ \varphi \end{array} \iff h_v^{(m)} = h_{\phi_v(v)}^{(m)} \quad \forall v \in V(G) \text{ and } 1 \le m \le 2d. \quad (58)$$

Because the graph readout function READOUT is injective, we obtain

$$h_v^{(m)} = h_{\phi_v(v)}^{(m)} \quad \forall v \in V(G) \text{ and } 1 \le m \le 2d \iff h_G = h_H. \quad (59)$$

Combining the two equations above proves the theorem. □

*Remark* A.27. One can in fact use Theorem 3.3 to establish a notion of weak isomorphism between a pair of graphs $G_1$ and $G_2$. Let $\mathcal{G}$ be the collection of finite connected undirected graphs. One may define a weak equivalence relation among the elements of $\mathcal{G}$ by constructing a function $F : \mathcal{G} \to \mathcal{X}$ from $\mathcal{G}$ to the space of regular cell complexes $\mathcal{X}$ with the property that two graphs $G_1$ and $G_2$ are weakly isomorphic if and only if their images $F(G_1)$ and $F(G_2)$ are isomorphic as cell complexes. We refer to Definition 8 of Bodnar et al. (2021a) for a related concept named "cellular lifting map". Theorem 3.3 proves that conventional GNNs establishes a weak isomorphism among graphs in $\mathcal{G}$ using the lifting function $F : \mathcal{G} \to \mathcal{X}_{univ}$ from $\mathcal{G}$ to the collection of universal covers of 1-dimensional cell complexes $\mathcal{X}_{univ}$.

## A.5 PROOF OF THEOREM 4.3

We now prove that the Cycle-to-Clique graph networks are more powerful than graph neural networks in distinguishing non-isomorphism classes of graphs.

*Remark* A.28. As stated in the main manuscript, there are four approaches that enrich the input data structure of graphs the graph neural network may process to distinguish non-isomorphic classes of graphs.

1. **Unique assignment of node labels**: We may distinguish distinct classes of graphs by assigning unique (or randomized) choices of node labels. Practical implementations include assigning node degrees as labels, imposing positional encoding, and implementing labeling tricks.

2. **Persistent homological methods**: We may incorporate homological invariants of graphs to GNNs by utilizing or constructing height functions over nodes and edges and constructing the associated persistence diagrams. These diagrams allow one to keep track of variations in the number of components and cycles throughout the filtration of graph dataset induced from the choice of the height functions.

3. **Topological Enrichment**: We may enrich topological structures of graphs by attaching simplicial and cellular complexes, or incorporating subgraph structures. Such actions allows conventional GNNs to encapsulate particular topological substructures of graphs suitable for classifying graph datasets of our interest, such as cliques within social media datasets or cyclic subgraphs within molecular datasets.

4. **Construction of non-isomorphic universal covers**: We may incorporate homological invariants of graphs, in particular cyclic subgraphs of graphs, to GNNs by constructing a new graph structure induced from subgraphs isomorphic to cyclic graphs. It is advised that the universal covers of newly induced subgraphs are not isomorphic to each other, thereby allowing the graph neural network to represent a collection of cyclic subgraphs as distinct vector representations.

**Definition A.29.** Denote by $C_n$ the cyclic undirected graph without self-loops consisting of $n$ nodes, and $K_n$ the undirected complete graph with $n$ nodes without self-loops.

We state a simple result that the induced universal covers of $C_n$'s are isomorphic, whereas those of $K_n$'s are not.

**Lemma A.30** (Lemma 4.2). *For any $n \neq m$, $\tilde{C_n}'' \cong \tilde{C_m}''$, whereas $\tilde{K_n}'' \not\cong \tilde{K_m}''$.*

*Proof.* The proof follows from the observation that any induced universal cover $\tilde{G}''$ of a k-regular graph $G$ is isomorphic to a directed infinite $2k + 2$ regular tree, where each node $v$ is connected by $k + 1$ directed edges from source nodes to $v$, and by $k + 1$ directed edges from $v$ to target nodes. $\square$

Hence, the lemma suggests a natural procedure to allow graph neural networks to distinguish cyclic graphs with distinct number of nodes: Substitute the cyclic graphs $C_n$ with complete graphs $K_n$. Using this observation, we may construct the Cy2C-GNN, as indicated in the main manuscript.

Before we recall the definitions of clique adjacency matrix and the architecture of the Cy2C-GNN, we first state the definition of a cycle basis $\mathcal{B}_G$ of a graph $G$.

**Definition A.31.** Let $G := (V, E)$ be a finite graph. A cycle basis $\mathcal{B}_G$ is a set of cyclic subgraphs of $G$ such that every cyclic subgraph of $G$ can be represented by unions and complements of elements in $\mathcal{B}_G$.

Given that $G$ is connected, a canonical method to construct a cycle basis of a graph $G$ is to find a spanning tree $T \subset G$. It is a non-trivial result from algebraic topology that if $G$ has $n$ nodes and $m$ edges, then the cardinality of any cycle basis of $G$ is equal to $m - n + 1$, see Theorem 2.44 of Hatcher (2002) for instance. Because $T$ is a subgraph of $G$ consisting of $n$ nodes and $n - 1$ edges, we immediately obtain that the remaining $m - n + 1$ edges of $G$ not lying in $T$ generates the elements of a cycle basis. Indeed, one can construct a cyclic subgraph by adding one of the $m - n + 1$ edges to the spanning tree $T$. Figure 10 illustrates an example how one can obtain a cycle

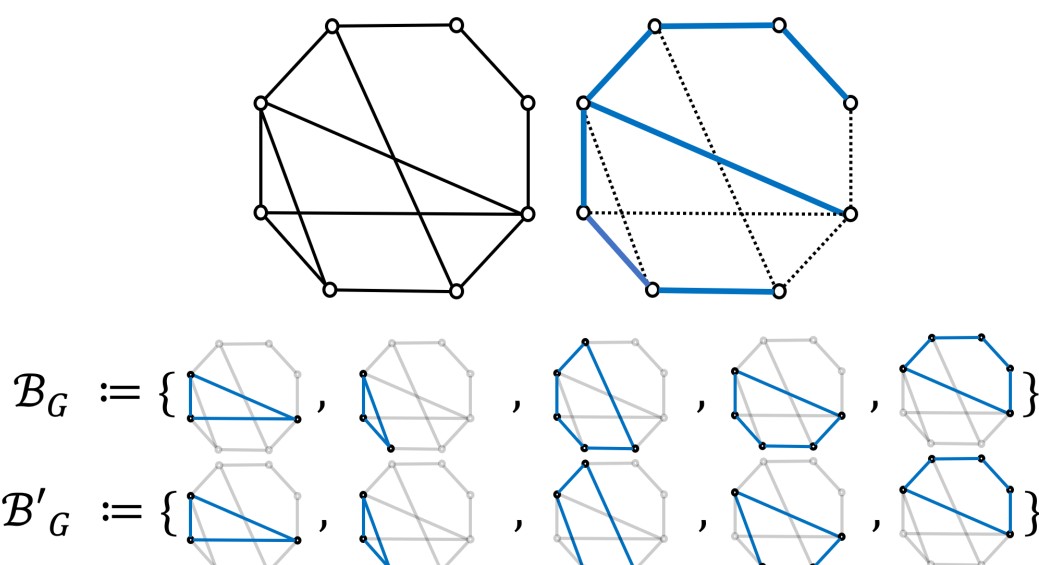

Figure 10: An illustration which shows how a choice of a spanning tree of a graph gives rise to a cycle basis $\mathcal{B}_G$ of a graph (Definition A.31). We note that the first, second, and the fifth cyclic subgraphs are chordless, whereas the third and the fourth cyclic subgraphs are not (Definition A.34). The non-chordless cyclic subgraphs can be further decomposed into chordless cyclic subgraphs, as shown in the new cycle basis $\mathcal{B}'_G$. (Theorem 4.3)

basis associated to a choice of a spanning tree $T$. There are systematic protocols to obtain cycle bases of graphs by conducting a standardized procedure in constructing spanning trees of a given graph $G$, such as Paton's algorithm which constructs spanning trees using the first or the last element methods Paton (1969), and persistent homological techniques applied over the set of edges $E(G)$ with a predetermined total order Hofer et al. (2020).

**Definition A.32** (Clique Adjacency Matrix (Definition 4.2)). Let $G := (V, E)$ be an undirected graph. Fix the cycle basis $\mathcal{B}_G$ of $G$, the set of cyclic subgraphs of $G$ which forms the basis of the cycle space (or the first homology group) of $G$. The clique adjacency matrix of $G$, denoted as $A_C$, is the adjacency matrix of the union of $\#\mathcal{B}_G$ complete subgraphs, each obtained from adding all possible edges among the set of nodes of each basis element $B \in \mathcal{B}_G$. Explicitly, the matrix $A_C := \{a_{u,v}^C\}_{u,v \in V(G)}$ is given by

$$a_{u,v}^C := \begin{cases} 1 & \text{if } \exists\, B \in \mathcal{B}_G \text{ cyclic s.t. } u, v \in V(B) \\ 0 & \text{otherwise} \end{cases} \tag{60}$$

Given $3 \le c_1 \le c_2 < \infty$, one may also define the bounded clique adjacency matrix $A_C|_{[c_1, c_2]} := \{a_{u,v}^C|_{[c_1, c_2]}\}_{u,v \in V(G)}$ which only substitutes cycles of size between $c_1$ and $c_2$ to cliques:

$$a_{u,v}^C|_{[c_1, c_2]} := \begin{cases} 1 & \text{if } \exists\, B \in \mathcal{B}_G \text{ cyclic s.t. } u, v \in V(B),\ c_1 \le |V(B)| \le c_2 \\ 0 & \text{otherwise} \end{cases} \tag{61}$$

For each node $v \in V(G)$, we denote by $\mathcal{C}(v)$ the set of nodes $w \in V(G)$ such that there exists a cyclic subgraph $C \in \mathcal{B}_G$ such that both $v$ and $w$ lie in $C$:

$$\mathcal{C}(v) := \{w \in V(G) \mid \exists\, C \in \mathcal{B}_G \text{ s.t. } v, w \in V(C)\} \tag{62}$$

**Definition A.33** (Cy2C-GNN). Let $G := (V, E)$ be a graph with $n$ nodes and continuous node attributes $f_G : V(G) \to \mathbb{R}^k$. We denote by Cy2C-GNN$^l$ the cycle-to-clique graph neural network comprised of a disjoint pair of two types of layers: A single neighborhood aggregating layer $H_C^{(1)}$ utilizing the clique adjacency matrix $A_C$: And $l$ layers of conventional neighborhood aggregating layers $H^{(m)}$ for $1 \le m \le l$ utilizing the adjacnecy matrix $A$. The Cy2C-GNN model admits the following three matrices as inputs:

- The adjacency matrix of a graph $A \in \mathbb{R}^{n \times n}$

- The node feature matrix $X \in \mathbb{R}^{n \times k}$ obtained from the function $f_G$

- The clique adjacency matrix $A_C \in R^{n \times n}$ (possibly bounded) from Definition 4.2.

Each types of message passing layer is constructed in the following manner:

1. The first layer $H_C^{(1)}$ of the network constructs the hidden node attribute $c_v^{(1)}$ using the clique adjacency matrix $A_C$ and the following composition of functions:

$$c_v^{(1)} := \text{COMBINE}^{(1)} \left( X_v, \text{AGGREGATE}_v^{(1)} \left( \{\{X_u \mid u \in \mathcal{C}(v)\}\} \right) \right) \qquad (63)$$

As for the other type of the message passing layer, the $m$-th layer $H^{(m)}$ of the network for each $1 \leq m \leq l$ constructs the hidden node attribute of dimension $k_m$, denoted as $h_v^{(m)}$ using the following composition of functions:

$$\begin{cases} h_v^{(m)} := \text{COMBINE}^{(m)} \left( h_v^{(m-1)}, \text{AGGREGATE}_v^{(m)} \left( \left\{ \left\{ h_u^{(m-1)} \mid \substack{u \in V(G), u \neq v \\ (u,v) \in E(G)} \right\} \right\} \right) \right) \\ h_v^{(0)} = X_v \end{cases}$$

$$(64)$$

For the two equations above, $X_v$ is the initial node attribute at $v$, $M_v^{(m)}$ is the collection of all multisets of $k_{m-1}$-dimensional real vectors with $\deg v$ elements counting multiplicities, the aggregation function

$$\text{AGGREGATE}_v^{(m)} : M_v^{(m)} \to \mathbb{R}^{k_m'} \qquad (65)$$

is a set theoretic function of $k_m'$-dimensional real vectors defined over $M_v$, and the combination function

$$\text{COMBINE}^{(m)} : \mathbb{R}^{k_{m-1}+k_m'} \to \mathbb{R}^{k_m} \qquad (66)$$

is a set theoretic function combining the attribute $h_v^{m-1}$ and the image of $\text{AGGREGATE}_v^{(m)}$. For obtaining the hidden attributes, identical aggregation and combination functions are employed.

2. Denote by $H^{(l)}$ be the final conventional neighborhood aggregating layer of the network.

- The final hidden attribute $H_v$ at node $v \in V(G)$ obtained from Cy2C-GNN$^l$ is obtained by concatenating the hidden attributes $c_v^{(1)}$ and $h_v^{(l)}$ at each $v \in V(G)$ and composing with multi-layer perceptrons (MLPs):

$$H_v = \text{MLP}(\text{CONCAT}(c_v^{(1)}, \ h_v^{(l)})). \qquad (67)$$

- For $1 \leq m \leq l$, let $M^{(m)}$ be the collection of all multisets of $k_m$-dimensional vectors with $\#V(G)$ elements. Let

$$\text{READOUT}^{(m)} : M^{(m)} \to \mathbb{R}^K \qquad (68)$$

be the graph readout function of $K$-dimensional real vectors defined over the multiset $M^{(m)}$. Then the vector representation of $G$, denoted as $H_{h_G, c_G}$, is given by

$$\begin{aligned} H_{h_G, c_G} &= \text{MLP}(\text{CONCAT}(H_{h_G}, \ H_{c_G})) \\ H_{h_G} &= \text{READOUT}^{(l)}(\{\{h_v^{(l)} \mid v \in V(G)\}\}) \\ H_{c_G} &= \text{READOUT}^{(1)}(\{\{c_v^{(1)} \mid v \in V(G)\}\}) \end{aligned} \qquad (69)$$

We end this section with the statement that Cy2C-GNN is more powerful than graph neural networks satisfying the conditions of Theorem 3.3 in distinguishing non-isomorphic classes of graphs.

**Definition A.34** (Chordless Subgraphs). Let $H \subset G$ be a subgraph. We say that $H$ is chordless if there does not exist a cyclic subgraph $C$ of $G$ such that $V(C) \subsetneq V(H)$ (See Figure 10 for examples of cyclic subgraphs which are chordless and not).

**Theorem A.35** (Theorem 4.3). *Let $G$ and $H$ be graphs which have isomorphic universal covers, endowed with node features $f_G : V(G) \to \mathbb{R}^k$ and $f_H : V(G) \to \mathbb{R}^k$. Fix a cycle basis $\mathcal{B}_G$ of $G$. Suppose that there exists a chordless cyclic subgraph $C \in \mathcal{B}_G$ such that any cycle basis $\mathcal{B}_H$ does not have any chordless cyclic subgraph of size equal to $|V(C)|$. Then Cy2C-GNN which utilizes bounded clique adjacency matrices can distinguish $G$ and $H$, whereas conventional GNNs cannot.*

*Proof.* Let $G$ and $H$ be graphs satisfying the conditions of the theorem. Let $C \in \mathcal{B}_G$ be the chordless cyclic subgraph of our interest. If a cyclic subgraph $C_H \in \mathcal{B}_H$ has size equal to $|V(C)|$, then $C_H$ is not chordless, i.e. there exist a set of chordless subgraphs $C_{H,i} \subset H$ such that $C_H = \cup_i C_{H,i}$. Because $C_H \in \mathcal{B}_H$, we may substitute the element $C_H$ with one of the $C_{H,i}$'s such that $|V(C_{H,i})| < |V(C_H)| = |V(C)|$. Therefore, we can assume that the cycle basis $\mathcal{B}_H$ does not contain any elements whose size is equal to $|V(C)|$. A visual illustration of the aforementioned argumentation can be found in Figure 10, where one can obtain a new cycle basis comprised of chordless cyclic subgraphs of strictly smaller sizes using the elements from a given cycle basis. We can hence apply a single layer of Cy2C-GNN equipped with the bounded clique adjacency matrix $A_C^{[c_1, c_2]}$ where $c_1 = c_2 = |V(C)|$. This results in transforming $G$ to be a non-trivial graph including the graph $C$, and transforming $H$ to be an empty graph. It is easy to see that the universal covers of such two graphs are not isomorphic to each other. $\square$

We note that one may still use the usual clique adjacency matrix to distinguish some of the pairs of graphs $G$ and $H$ satisfying the conditions of Theorem 4.3. One needs to verify, however, that the resulting induced graphs associated to clique adjacency matrices of $G$ and $H$ have non-isomorphic universal covers. The upcoming example gives exemplary pairs of graphs for every node $|V(G)| \geq 6$ which are discernible by using Cy2C-GNN equipped with the usual clique adjacency matrix.

*Example A.36.* In this example, we explicitly construct a collection of isomorphism classes of graphs with $n$ nodes such that Cy2C-GNN can distinguish, whereas conventional GNNs cannot.

Consider the two connected graphs $G$ and $H$ with 6 nodes as shown in Figure 11. Theorem 3.3 implies that any GNN (including the Weisfeiler-Lehman test) cannot distinguish the two graphs because the induced initial node attributes over the universal covers $\tilde{G}''$ and $\tilde{H}''$ are equal.

In fact, for any even number of nodes $2n \geq 6$, there exists a pair of two labelled connected graphs with $2n$ nodes that any GNNs cannot distinguish. Consider a graph $G_n$ constructed from gluing two $C_{n+1}$ cyclic graphs along a distinguished edge. Consider another graph $H_n$ constructed from connecting a pair of distinguished nodes from two disjoint $C_n$ cyclic graphs by an edge. Impose identical attributes to nodes based on their degrees. Note that there are 2 nodes of degree 3 and $2n - 2$ nodes of degree 2. Then the two universal covers $\tilde{G}_n''$ and $\tilde{H}_n''$ are isomorphic. Furthermore, the node attributes over the universal covers induced from $G_n$ and $H_n$ are identical. Theorem 3.3 hence implies that any GNNs cannot distinguish $G_n$ and $H_n$. However, the two graphs are clearly not isomorphic.

Let $G_{n,1}$ be the graph obtained by substituting the two cyclic subgraphs $C_{n+1}$ with two complete graphs $K_{n+1}$. Likewise, let $H_{n,1}$ be the graph obtained by substituting the two disjoint cyclic graphs $C_n$ with two complete graphs $K_n$. Among the nodes of $G_{n,1}$, there are $2n - 2$ nodes of degree $n$ and 2 nodes of degree $n + 2$. As for the nodes of $H_{n,1}$, there are $2n - 2$ nodes of degree $n - 1$ and 2 nodes of degree $n$. Thus, $\tilde{G}_{n,1}'' \not\cong \tilde{H}_{n,1}''$ (In fact, $\tilde{H}_{n,1}''$ consists of two disjoint isomorphic copies of infinite trees, each of which is not isomorphic to $\tilde{G}_{n,1}''$). Therefore, any GNNs satsifying the conditions of Theorem 3.3 may distinguish the graphs $G_{n,1}$ and $H_{n,1}$. The Cy2C-GNN captures the non-isomorphism of universal covers by admitting the clique adjacency matrices of $G$ and $H$ as inputs. Note that the clique adjacency matrix of $G$ (and $H$) is equal to the sum of the adjacency matrix of $G_{n,1}$ (and $H_{n,1}$, respectively) and the identity matrix.

Likewise, there are a collection of finite graphs with many connected components that any graph neural network cannot distinguish. Consider a collection of graphs $\mathcal{G}$ with $n$ nodes consisting of isomorphism classes of disjoint union of cyclic graphs $\bigsqcup_{\substack{i \in I \\ \sum m_i = n}} C_{m_i}$. See Figure 12 for instance.

The directed unfolding tree $T_v^l$ of depth $l$ at each node $v$ of the graph is isomorphic to a directed 3-regular rooted tree of depth $l$. If the node attributes are constant, then Theorem 3.3 implies that any graph neural networks cannot distinguish all such disjoint union of cyclic graphs.

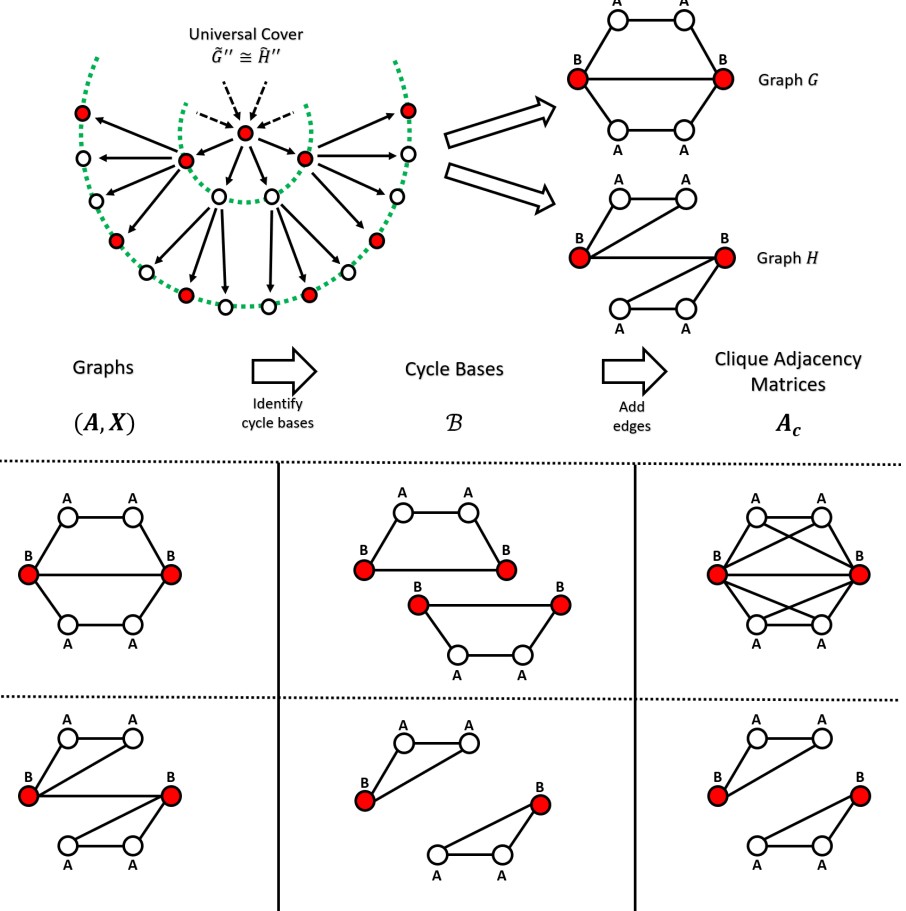

Figure 11: An illustration of Definition 4.2 and Theorem 4.3. The two connected graphs have identical associated universal covers, which implies that graph neural networks cannot distinguish the two graphs. Nevertheless, Cy2CGN is able to distinguish them by adding complete graphs from cycles of these graphs, whose universal covers are not isomorphic. The incorporation of cycle bases of graphs to graph representations become feasible via admitting the clique adjacency matrices as inputs.

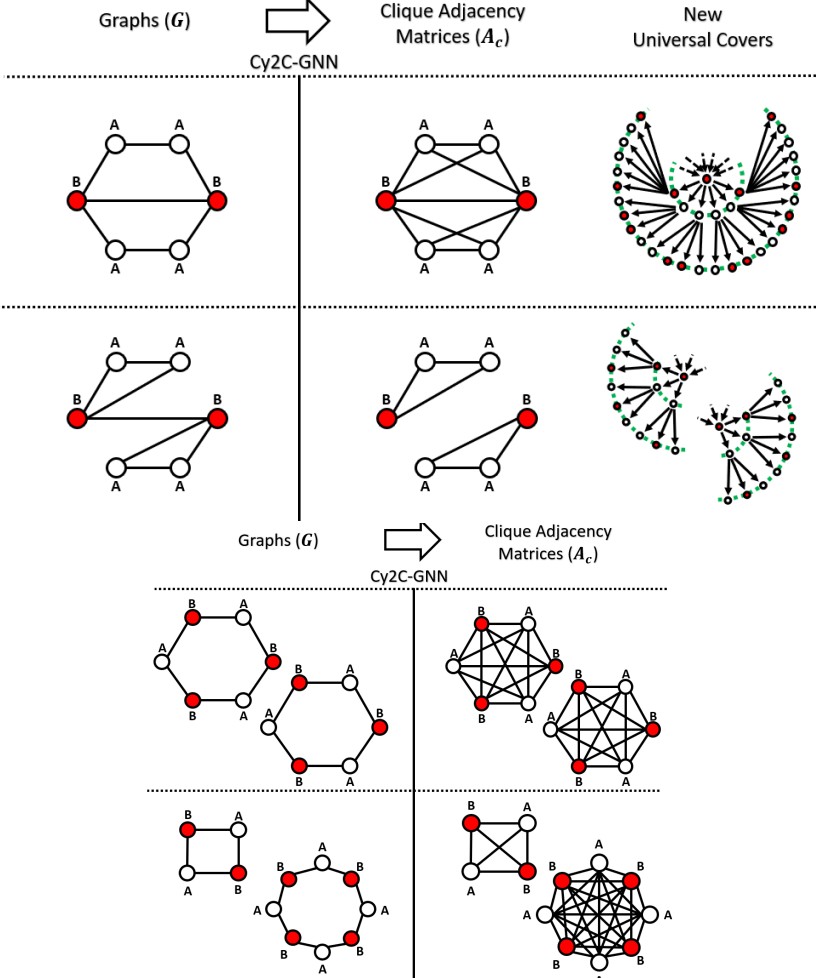

Figure 12: An illustration of Theorem 4.3. Each of the following pair of graphs with non-isomorphic cyclic subgraphs have identical associated universal covers, which implies that graph neural networks cannot distinguish the two graphs. Nevertheless, Cy2C-GNN is able to distinguish them by adding complete graphs from cycles of these graphs, whose universal covers are not isomorphic.

By substituting every disjoint cyclic graphs with complete graphs, however, we obtain non-isomorphic classes of disjoint unions of universal covers. Therefore, even if the node attributes are constant, Theorem 3.3 implies that Cy2C-GNN can distniguish all such disjoint union of cyclic graphs.

*Remark* A.37. One immediate result from Theorem 4.3 is that Cy2C-GNN can distinguish non-isomorphism classes of graphs that 1,2,and 3-WL tests cannot distinguish. The proof of this claim originates from the adaptation of the well-known arguments presented in Bodnar et al. (2021b;a). The Rook's $4 \times 4$ graph and the Shrikhande graph, which constitutes the only two elements of the strongly regular graphs in family SR(16,6,2,2), are not isomorphic because the Shrikhande graph contains a chordless cyclic subgraph of size 5, whereas Rook's $4 \times 4$ graph does not, see for instance Table 4 of Bodnar et al. (2021a). Theorem 4.3 hence implies that Cy2C-GNN can distinguish these two graphs by incorporating the cycle basis of the Shrikhande graph which contains a chordless cyclic subgraph of size 5, and that of the Rook's $4 \times 4$ graph which does not contain such subgraphs.

We note that contemporary techniques which are known to be able to classify classes of graphs that 1,2 and 3-WL tests cannot possess capability in counting the number of nodes of a given cyclic or clique subgraphs of $G$. Such results include Proposition 25 of Bodnar et al. (2021a), Proposition 2 of You et al. (2021), Theorem 8 of Bodnar et al. (2021b), and Proposition 3.4 of Bouritsas et al. (2022).

While these previously studied GNNs and Cy2C-GNN share a common objective in distinguishing cyclic substructures of graphs, the inherent algorithm which manifests such objectives are different. To elaborate, Bodnar et al. (2021a) and Bouritsas et al. (2022) propose gluing higher dimensional cells or complexes while preserving the 1-skeleton structure of a given graph. You et al. (2021) proposes perturbing the feature of a designated node, which allows GNNs with sufficiently large number of layers to distinguish cyclic substructures of graphs. Cy2C-GNN, on the other hand, circumvents utilizing higher dimensional cells or perturbing the given node attributes by adding suitable edges, or transforming the 1-skeleton structure of a graph. This procedure allows a wide range of classes of graphs, in particular those with varying cyclic structures, with isomorphic universal covers to attain additional non-isomorphic universal covers. Because neighborhood aggregating layers are effective in distinguishing isomorphism classes of graphs up to their universal covers, Cy2C-GNN allows conventional GNNs to detect such cyclic structures with a single neighborhood aggregating layer utilizing clique adjacency matrices.

## A.6    Computational Complexity

The computation complexity of Cy2C-GNN method consists of two parts: An extraction algorithm for extracting clique adjacency matrix from general description of graph: And the GNN model with clique adjacency matrix. The extraction algorithm is equivalent to constructing a basis of cycles of the first homology group of a given graph $G$. The classical algorithm proposed by Paton proves that the time complexity of finding a cycle basis of a finite graph with $n$ nodes is given by $O(n^\gamma)$ for $2 \leq \gamma \leq 3$ Paton (1969). For random graphs with $n$ nodes and density $0.5$, the time complexity of Paton's algorithm is equal to $O(n^2)$. Indeed, as shown in Table 3 in Appendix B, the average sizes of cycle bases $\mathcal{B}_G$ for benchmark graph datasets are bounded above by their average numbers of edges. There are other more efficient algorithms than Paton's algorithm in obtaining a cycle basis of a finite graph with $n$ nodes. One may resort to using persistent homological techniques to construct persistent diagrams associated to filtrations of a finite graph $G$ with $m$ edges, whose time complexity is of order $O(m\alpha(m))$ Horn et al. (2021). The function $\alpha(m)$ is the inverse Ackermann function, which grows at an extremely slow rate in terms of $m$. In fact, it grows at such a slow rate that we may even assume without loss of generality that $\alpha(m)$ is a constant in terms of $m$ for any practical choice of the number of edges $m$.

The latter architectural component of the Cy2C-GNN algorithm shares equivalent time complexities to conventional GNNs, corresponding to $O(m_C + lm_1)$, where $l$ is the number of layers, $m_C$ is the number of edges of a graph associated to the clique adjacency matrix $A_C$, and $m$ is the number of edges of $G$. We recall that the Euler characteristic formula implies

$$\#\mathcal{B}_G = \#E(G) - \#V(G) + 1$$

for any connected graph $G$. Hence, the time complexity of the Cy2C-GNN method is practically equivalent to $O(m)$, given that the number of layers is fixed and the number of nodes consisting each cycle bases is bounded.

We established that as long as one chooses a sufficiently efficient algorithm for computing cycle bases of graphs, the time complexity to represent graphs with $n$ nodes and $m$ edges using Cy2C-GNN is equivalent to $O(m)$. Such a complexity is equivalent to that of conventional GNNs, and more efficient than persistent homological techniques, while managing to capture cyclic substructures of graph data sets. The time complexity of constructing persistence diagrams given an arbitrary height function over the set of nodes and edges of a graph $G$ is equal to $O(n^{2w})$, where $w$ is a positive number such that $O(n^w)$ is the time complexity required for multiplying two $n \times n$ matrix Milosavljevič et al. (2011). We note that the time complexity of conventional matrix multiplication algorithm is given by $O(n^3)$, with the best asymptotic complexity running in $O(n^w)$ with $w \sim 2.37$ Alman & Williams (2021). As for persistent homological techniques obtained from a predetermined height function over $G$, there are classes of height functions whose time complexity to construct persistence diagrams is equal to $O(n^2 \log n)$, see for instance the work by Rieck et al Rieck et al. (2019). Nevertheless, such persistent homological techniques may not necessarily distinguish all isomorphism classes of graphs, as carefully studied in the work by Horn et al Horn et al. (2021). For example, consider the height function $h : G \to \mathbb{R}$ over a graph $G$ with continuous node attributes $f_G : V(G) \to \mathbb{R}^k$ defined as

$$\begin{cases} h(v) = 0 \text{ for } v \in V(G) \\ h(e) = \|f_G(v) - f_G(w)\|_p + \tau \text{ for } e = (v, w) \in E(G) \end{cases} \tag{70}$$

where $\| \cdot \|_p$ is the $L_p$-distance defined over $\mathbb{R}^k$, and $\tau > 0$ is a positive bias term. Let $G$ and $H$ be two graphs as shown in Figure 4. Then one may observe that there exists a bijection between the zeroth dimensional persistence diagrams (and first dimensional persistence diagrams, respectively) obtained from filtrations of $G$ and $H$ with respect to the height function $h$.

## B  IMPLEMENTATION

The experiments were run on cloud provider, comprising of 16 physical cores (Intel Xeon Processor (Skylake, IBRS) CPU processor @ 2.10GHz) with 2 NVIDIA Quadro RTX 6000 GPUs.

### B.1  EXPERIMENTAL SETUP

**Dataset** We perform graph classification on the 3 bioinformatics(DD, PROTEINS(FULL), EN-ZYMES), 3 social network datasets (COLLAB, IMDB-B, REDDIT-B ), 3 small molecules datasets ( MUTAG, NCI1, NCI109 ), 3 datasets with edge features (BZR-MD, COX2-MD, PTC-MR) and 4 large datasets (REDDIT-M-5K, MOLHIV, MOLTOX21, MOLTOXCAST) from TU datasets Morris et al. (2020) which are available in *pytorch-geometric* library Fey & Lenssen (2019) and Open Graph Benchmark datasets Hu et al. (2021). We use one-hot encoding for discrete node features and normalize the continuous node features before using GNNs. We make initial node features for the social dataset by utilizing one-hot encoding of node degrees. The details of statistical properties of benchmark data sets are summarized in Table 3 and 4. For ablataion study, we perform an ablation study by utilizing the "CYCLES" and "NECKLACES" synthetic datasets from Horn et al. (2021).

**Models** We use three baseline models to compare the proposed methods comprised of GCN, GAT, and GIN. The hidden dimension of each message passing layer is uniformly given by 136. The number of heads in GAT is equal to 8, and each hidden size is equal to 17 for every benchmark dataset. For implementing GCN and GAT, the element-wise mean pooling layer is used for constructing the outputs of the final message passing layer. We use a classifier that consists of three MLPs for classifying graph data sets. In the case of GIN, each layer passes the hidden attributes through the element-wise sum pooling. Furthermore, summed graph representations of each layer are also summed after passing MLPs to obtain the entire graph representation.

Cy2C-GNNs share identical architectural structures to GCN, GAT, and GIN, except for the additional layer for clique adjacency matrix and the dimensions of the hidden state at the first MLP in the classifier. All baseline models and Cy2C-GNNs include batch normalization Ioffe & Szegedy (2015) and residual connection He et al. (2016). The hyperparameters of Cy2C-GNNs are optimized with hidden dimensions from 32 to 256, weight decay from 0.0 to 0.01, the number of message passing layer from 1 to 5, initial dropout rate from 0 to 0.8, dropout rate of message passing layer from 0 to 0.8 and the number of layer in classifier from 1 to 3. In case of large data sets, we implement Cy2C-GNN models whose number of layers are between 1 and 7. We also consider both methods for classifier

Table 3: A summary of statistics of bioinformatics, social network graph and small molecules datasets. Cells notated as "-" indicate graph data set which do not have or use features in this paper. Numbers in parentheses are dimension of features. The term "Average # H1 Cycles" indicates the average size of the cycle bases of graphs. The term "Average Magnitude # Cycles" denotes the average number of nodes present in a cyclic subgraph of a graph.

| | BIOINFORMATICS | | | SOCIAL NETWORK | | | SMALL MOLECULES | | |
|---|---|---|---|---|---|---|---|---|---|
| DATA SETS | D&D | PROTEINS(FULL) | ENZYMES | COLLAB | IMDB-B | REDDIT-B | MUTAG | NCI1 | NCI109 |
| # GRAPHS | 1178 | 1113 | 600 | 5000 | 1000 | 2000 | 188 | 4110 | 4127 |
| # CLASS | 2 | 2 | 6 | 3 | 2 | 2 | 2 | 2 | 2 |
| AVERAGE # NODES | 284.32 | 39.06 | 32.63 | 74.49 | 19.77 | 429.63 | 17.93 | 29.87 | 29.68 |
| AVERAGE # EDGES | 715.66 | 72.82 | 62.14 | 2457.78 | 96.53 | 497.75 | 19.79 | 32.30 | 32.13 |
| AVERAGE # H1 CYCLES | 432.36 | 34.84 | 30.75 | 2383.72 | 77.76 | 70.61 | 2.86 | 3.62 | 3.64 |
| AVERAGE MAGNITUDE # CYCLES | 7.21 | 3.72 | 3.69 | 3.04 | 3.00 | 5.412 | 6.24 | 5.94 | 5.92 |
| # GRAPH WITHOUT CYCLES | 0 | 1 | 3 | 0 | 0 | 67 | 0 | 83 | 97 |
| NODE FEATURES DIMENSION | 89 | 29 | 18 | - | - | - | 7 | 37 | 38 |

Table 4: A summary of statistics for graph classification datasets with edge features and large datasets including OGB. Numbers in parentheses are dimension of features. The term "Average # H1 Cycles" indicates the average size of the cycle bases of graphs. The term "Average Magnitude # Cycles" denotes the average number of nodes present in a cyclic subgraph of a graph.

| | DATASETS WITH EDGE FEATURES | | | LARGE DATASETS | | | |
|---|---|---|---|---|---|---|---|
| DATA SETS | BZR-MD | COX2-MD | PTC-MR | REDDIT-M-5K | MOLHIV | MOLTOX21 | MOLTOXCAST |
| # GRAPHS | 306 | 303 | 344 | 4999 | 41,127 | 7,831 | 8,576 |
| # CLASS | 2 | 2 | 2 | 5 | 2 | 2 | 2 |
| AVERAGE # NODES | 21.30 | 26.28 | 14.29 | 508.52 | 25.5 | 18.6 | 18.8 |
| AVERAGE # EDGES | 225.06 | 335.12 | 14.69 | 594.87 | 27.5 | 19.3 | 19.3 |
| AVERAGE # H1 CYCLES | 204.75 | 309.84 | 1.40 | 90.08 | 3.03 | 1.75 | 1.75 |
| AVERAGE MAGNITUDE # CYCLES | 3.0 | 3 | 4.30 | 7.73 | 5.65 | 4.55 | 4.53 |
| # GRAPH WITHOUT CYCLES | 0 | 0 | 90 | 51 | 1582 | 1781 | 1956 |
| NODE FEATURES (DIM) | 8 | 7 | 18 | - | 9 | 9 | 9 |

with last layers outputs and with concatenated outputs of all layers. For classifying BZR-MD and COX2-MD datasets, we require that Cy2C-GNNs incorporate edge attributes of respective data sets. For classifying OGB and PTC-MR data sets, however, we implement both cases where Cy2C-GNNs incorporate the given edge features or not.

**Graph classification** For classifying graph data sets, we adapted stratified 10-fold cross-validations to evaluate the performance of baseline models and Cy2C-GNNs, while preserving the percentage of the train, validation, and test samples for each class. Theses models are optimized by Adam optimizer Kingma & Ba (2014) with a initial learning rate from $1 \times 10^{-4}$ to $1 \times 10^{-3}$. We use *ReduceLROnPlateau* for the learning rate scheduler in Pytorch library, which multiplies the learning rate by 0.8 when validation loss does not decrease with the patience of 25. Additionally, we use the early stopping criterion when validation loss does not decrease during the 100 epochs. Training sequences are stopped when the learning rate becomes below the minimum learning rate of $1 \times 10^{-6}$. In case of OGB datasets, we perform overall experiments 10 times with original training/validation/test dataset splits Hu et al. (2020) and batch size is set to 128. Cy2C-GNNs are also optimized by Adam optimizer Kingma & Ba (2014). *ReduceLROnPlateau* are used, which multiplies the learning rate by 0.8 whenever the validation accuracy does not increase during the 25 epochs, along with the stopping criterion of patience of 100.

## B.2 DETAILS OF DATASET

Tables 1 and 2 summarizes the statistical properties of benchmark data sets utilized in demonstrating the effectiveness of the proposed model.

The synthetic datasets "CYCLE" and "NECKLACE" are generated using the github repository provided from Horn et al. (2021). Each dataset consists of 1000 synthetically generated graphs with two distinct cyclic substructures, each class of which contains exactly half (for instance 500 graphs out of 1000 graphs) out of all generated graphs. The objective is to verify whether the proposed GNN can

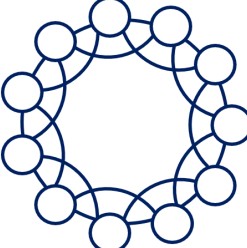 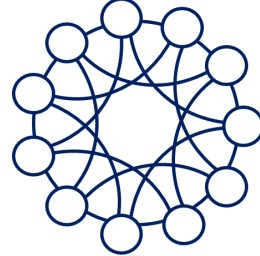

Figure 13: An example of CSL data Murphy et al. (2019) with skip length 2 and 3. WL test and Conventional GNNs cannot distinguish these graphs.

accurately implement binary classification by detecting differences in the proposed cyclic structures. The "CYCLE" dataset consists of either a connected cyclic graph with large number of nodes or a disconnected cyclic graphs with smaller number of nodes. The "NECKLACE" dataset consists of two classes of graphs $\mathcal{B}_G$ satisfying $|\mathcal{B}_G| = 2$ such that either the two elements $C_1, C_2 \in \mathcal{B}_G$ have a non-trivial intersection or are disjoint. A graphical description of the two datasets can be found in Figure 1 of Horn et al. (2021).

### B.3  DETAILS OF IMPLEMENTATION

**Extraction algorithm** We used the following extraction algorithm to construct the clique adjacency matrix. First, we use the function *cycle_basis* Paton (1969) from *NetworkX* library Hagberg et al. (2008) which returns a list of cycle basis elements of a graph. The node sets of cycle basis are used for making new edges that transform each cycle to a clique. Then, the clique adjacency matrix is obtained by masking edges that do not include any cycle and adding new edges from the adjacency matrix. We refer to Figure 11 to recall how the clique adjacency matrix is constructed.

**Running time analysis** We performed additional experiments on measuring the running time of the baseline GNNs and Cy2C-GNNs, including the pre-processing steps required for Cy2C-GNNs, such as constructing clique adjacency matrices and detecting cycle bases. Since the REDDIT-M-5K dataset has the highest average number of nodes and edges in our benchmark datasets, we selected the dataset to compare additional experiments. First, the CPU's running time for preprocessing steps takes an average of 0.49 seconds for each graph and 2461.44 seconds for all graphs in the dataset. For a fair comparison, we evaluate the GPU's running time of baseline GCNs and Cy2C-GCNs with 128 hidden dimensions in the train sequence. We perform five iterations of implementing GCNs and Cy2C-GCNs with identical hyperparameters, and measure the average time spent for reaching the same number of epochs to analyze additional computational costs derived from the clique adjacency matrix. Table 5 shows the running time of baseline GCNs and Cy2C-GCNs with different numbers of layers. Cy2C-GNNs with a single message passing layer takes approximately 1.5 times more time to represent graphs compared to baseline GCNs with the same number of layers. Nevertheless, we observe that increasing the number of layers can significantly decrease the relative computational costs of Cy2C-GNNs to baseline GCNs. This observation is suggested from the fact that Cy2C-GNNs with five message passing layers takes approximately 1.2 times more time to represent graphs compared to baseline GCNs with identical architectural structures, a marked improvement in relative running time compared to the case where Cy2C-GNNs had a single message passing layer.

The running time of Cy2C-GNNs, including preprocessing steps, is significantly longer than that of the baseline GNNs. However, since preprocessing steps need to be performed only once for the first time and can be parallelized, we can claim that Cy2C-GNNs have comparable computational complexity to baseline GCNs.

**Additional ablation studies** We performed an additional ablation study to empirically shows the expressive power of Cy2C-GNNs by comparing results of conventional GNNs, Relational Pooling GIN(RP-GIN) Murphy et al. (2019) and our method obtained from Circular Skip Link(CSL) dataset Murphy et al. (2019). Figure 13 shows the example of CSL data that have different skip length

Table 5: Comparison of running time (in seconds) at 50, 100 and 150 epoch obtained from REDDIT-M-5K.

| | BASELINE GCN-N | | | | |
|---|---|---|---|---|---|
| EPOCH | N=1 | N=2 | N=3 | N=4 | N=5 |
| 50 | 30.6 | 37.4 | 44.7 | 52.1 | 59.4 |
| 100 | 60.0 | 74.8 | 89.4 | 103.8 | 119.5 |
| 150 | 89.0 | 112.7 | 133.7 | 156.2 | 179.1 |
| | CY2C-GCN-N | | | | |
| EPOCH | N=1 | N=2 | N=3 | N=4 | N=5 |
| 50 | 43.2 | 50.1 | 58.4 | 65.4 | 72.5 |
| 100 | 85.7 | 100.6 | 115.8 | 131.3 | 145.4 |
| 150 | 129.2 | 150.7 | 173.3 | 196.7 | 217.8 |

Table 6: Classification resutls obtained from CSL dataset. Classification methods with grey color text are cited from results obtained from pre-existing publication Murphy et al. (2019). The term "Baseline GCN-N" denotes conventional GCNs with $N$ layers, where $N$ takes any value in $\{1, 2, 3, 4, 5\}$.

| | CSL DATASET | | |
|---|---|---|---|
| MODEL | ACCURACY | MAX | MIN |
| GIN | 10±0.0 | 10 | 10 |
| RP-GIN | 37.6±12.9 | 53.3 | 10 |
| BASELINE GCN-N | 10±0.0 | 10 | 10 |
| CY2C-GCN-1 | 91.3±1.6 | 93.3 | 90 |

$R$. CSL dataset consists of graphs where $R \in \{2, 3, 4, 5, 6, 9, 11, 12, 13, 16\}$ with 41 nodes that have the same node features. We evaluate baseline GCNs and Cy2C-GCN-1 with 5-fold cross-validations while preserving the percentage of the train and validation in reference Murphy et al. (2019). The hidden dimension of each message passing layer is fixed by 16, and the batch size is set to 16. Baseline GCNs and Cy2C-GCN-1 are optimized by Adam optimizer Kingma & Ba (2014) with a learning rate from $1 \times 10^{-4}$. Since only the train dataset and validation dataset exist, we choose the best validation accuracy in terms of the accuracy of the train dataset. The results are listed in Table 6. The results of baseline GCNs is consistent with the fact that the conventional GNNs cannot distinguish graphs in CSL dataset Murphy et al. (2019). Cy2C-GCN-1 not only distinguishes graphs in CSL dataset, but also shows much higher performance than the RP-GIN.

