# OpenReview forum: "Cycle to Clique (Cy2C) Graph Neural Network: A Sight to See beyond Neighborhood Aggregation"
_ICLR.cc/2023/Conference — ICLR 2023 poster_

### Official Review · Reviewer_L1b5 · 2022-10-24

**Confidence:** 3
**Correctness:** 4
**Technical Novelty And Significance:** 4
**Empirical Novelty And Significance:** 2
**Recommendation:** 8

**Clarity, Quality, Novelty And Reproducibility:**

Clarity is good. The authors can further improve the clarity of the paper by referring to where in the appendix the definition of certain graph theoretic term is provided. For example, cycle basis is defined in Definition A.31 and the author could refer to Definition A.31 when the term first appeared in the main text.

Quality is good, the empirical experiments section can be improved.

Originality is good. I think the idea that uses cliques from cyclic substructures of the graph to perform neighborhood aggregation is novel.

**Strength And Weaknesses:**

The paper is well motivated and nicely written. The authors provide solid theoretical justifications for the proposed Cy2C-GNN architecture. I liked the fact that this architecture is provably more powerful and simple at the same time, which makes it very accessible to practitioners. I also liked that the authors discuss the limitations of the architecture in the conclusion.

The weak part of this paper is the empirical evaluations. It seems that the authors are comparing the best performance between baseline models and Cy2C-GNN models. Based on the results reported in Table 1 and Table 2, it is difficult to understand the effect of the additional neighborhood aggregation layer that uses the clique adjacency matrix. For example, for PROTEINS(FULL) in Table 1, the reported baseline GNN is GAT-3 whereas the reported Cy2C-GNN is Cy2C-GCN-4. Both methods use different GNN architecture and also different number of layers (please correct me if I am wrong), therefore it is difficult the understand what exactly had led to the performance gain. The results would be more informative if the authors report results in the setting where the only architectural difference is the additional clique averaging layer (and the final layer of course).

**Summary Of The Paper:**

This paper proposes a new graph neural network architecture termed Cycle to Clique Graph Neural Network (Cy2C-GNN). The new architecture is simple in that it just adds a separate layer in standard graph neural networks, and thus it maintains the efficiency of simple GNNs such as GCN; at the same time, the new architecture is based on graph theoretic considerations and is provably more powerful than standard GNNs at distinguishing certain isomorphism classes of graphs. The authors also empirically demonstrate that Cy2C-GNN can efficiently and reliably represent cyclic structures of graph datasets. Over a range of benchmark datasets Cy2C-GNN achieves superior or comparable graph classification performance than the current state-of-the-art methods.

**Summary Of The Review:**

The paper proposes a simple and novel graph neural network architecture based on universal covering graphs. The new architecture is backed by theoretical insights and also it is shown empirically to achieve state-of-the-art graph classification performance. Even though I think the empirical evaluations are not good enough, overall the strengths of this paper oversights its weaknesses.

---

> ### Author Response · Authors · 2022-11-14
> **Response to Reviewer L1b5**
>
> We sincerely thank the reviewer for giving very positive and encouraging feedbacks, which are of great help for revising the manuscript.
>
> [Experiments]
>
> We sincerely thank the reviewer for giving a careful assessment on the empirical evaluations of the paper. To bolster the empirical results of Cy2C-GNN, we updated the manuscript as follows. For Table 1, we listed the best results, each obtained from baseline GCN, GAT, and GINs with message passing layers ranging between 1 and 5. We can observe that Cy2C-GNN enhances classification results obtained from three variants of baseline GNNs over all bioinformatics, social network, and small molecules datasets. We also added additional experiments in the appendix of the manuscript on distinguishing the synthetic dataset ``Circular Skip Link Dataset’’ (from “Relational pooling for graph representations” by Murphy, R., Srinivasan, B., Rao, V., and Ribeiro, B. (ICML, 2019)). The dataset was obtained from the respective github repository, and we compared the performances of Cy2C-GCN with 1 clique averaging layer to GCN with 1, 2, 3, 4, and 5 message passing layers, RP-GIN, and GIN from the reference. Given that the dataset is only divided into training and validation set, we apply the stopping criterion for only the training dataset. Experimental results show that using Cy2C-GCN with a single neighborhood aggregation layer significantly enhances the capability of baseline GNNs, RP-GIN, and GIN in distinguishing circular skip link datasets. One way to understand such stark difference in classification results rests upon how clique averaging layer is designed. The layer transforms cyclic subgraphs into complete subgraphs, and masks other edges which do not lie inside the cyclic subgraphs. We can deduce that such operations result in highlighting the geometric differences in cyclic substructures of circular skip link datasets. We hope this additional experiment shines light to how architectural difference originating from additional clique averaging layers makes a difference in distinguishing cyclic substructures of graphs.
>
> [Clarifications]
>
> As kindly suggested by the reviewer, we reorganized some of the technical mathematical formulations to the appendix of the paper and supplemented each definition and theorems with corresponding statements in the appendix. Further explanations on what these mathematical concepts represent are also added in the updated manuscript.

---

### Official Review · Reviewer_Qt5k · 2022-10-24

**Confidence:** 2
**Correctness:** 4
**Technical Novelty And Significance:** 3
**Empirical Novelty And Significance:** 2
**Recommendation:** 6

**Clarity, Quality, Novelty And Reproducibility:**

The paper is quite clear, and the theoretical results looks quite original. Practically speaking, the new proposed architecture is quite incremental though.

**Strength And Weaknesses:**

Strengths:
---The paper is well written and easy to follow
---The experiments are quite exhaustive
---To my knowledge, Theorem 3.3 is new and interesting

Weaknesses:
---The method sounds a bit complicated. It would be nice to better motivate the use of universal covers: as far as I understand, extended persistent homology would be enough to detect and process such cycles in separate layers. Thus, it would be nice to either discuss or compare more to this alternative.
---In terms of architecture design, the proposed neural net seems a bit incremental

**Summary Of The Paper:**

In this article, the authors propose to improve graph neural networks by characterizing and processing the graph cycles using universal covers, which is motivated by a characterization of the classes of graphs that usual graph neural nets can distinguish. More precisely, they provide a simple yet efficient procedure for turning cycles into cliques and for processing clique adjacency matrices with new graph neural network layer combined with standard aggregation schemes. They finally show improvements over competitors over several real-world data sets.

**Summary Of The Review:**

Overall, the paper looks good to me, even though I am no expert in graph neural networks.

---

> ### Author Response · Authors · 2022-11-14
> **Response to Reviewer Qt5k**
>
> We sincerely thank the reviewer for giving constructive feedbacks, which are of great help for revising the paper.
>
> [Comparison to Persistent Homology]
>
> This is an excellent question. The ablation studies conducted in the experiments section suggests that static persistent homological techniques, where one uses predetermined height functions for constructing persistent diagrams, is not effective enough to distinguish wide ranges of cyclic structures of graphs with node attributes. To circumvent this problem, one may need to implement dynamic persistent homological techniques, where GNNs construct suitable height functions for each graph datasets, as proposed in Rieck et al (2021). However, the computational complexity required for constructing dynamic persistent diagrams suitable for large graph datasets may be expensive.
>
> Cy2C-GNN avoids this problem by tackling the problem of representing cyclic substructures from a different angle, as suggested in the discussion at the start of Section 4 of the manuscript. Rather than representing homological properties of graphs, we focus on representing homotopic properties of graphs by changing the geometric structure of their universal covers. We would like to note that there is a correspondence (called the Galois correspondence) between covering spaces and subgroups of homotopy groups of topological spaces, see for instance Section 1.3 of Hatcher (2002). These homotopic properties of graphs harbor richer structures than homological properties of graphs. For example, as indicated in Section 2.A of Hatcher (2002), homology groups are quotients (in fact abelianizations) of homotopy groups of spaces. The crux of Cy2C-GNN and Theorem 3.3 is that one way to effectively represent homotopic properties of graphs is by adding additional edges to cyclic subgraphs, which transforms the universal covers of the original graphs, hence altering the subgroup structure of their homotopy groups (in accordance with the Galois correspondence). This hence allows Cy2C-GNNs to encapsulate cyclic structures of graphs without necessarily training height functions for constructing persistence diagrams, while ensuring comparable time complexity to baseline GNNs.
>
> To further strengthen the empirical effectiveness of Cy2C-GNNs, we added additional experiments in Appendix B of the manuscript on distinguishing the synthetic dataset ``Circular Skip Link Dataset’’ (from “Relational pooling for graph representations” by Murphy, R., Srinivasan, B., Rao, V., and Ribeiro, B. (ICML, 2019)) and comparing running times to baseline GNNs, as kindly requested by one of the other reviewers for our paper. Experimental results suggest that Cy2C-GCN with a single message passing layer is more effective than baseline GCNs in detecting cyclic substructures, while exhibiting comparative execution times.

---

> > ### Comment · Reviewer_Qt5k · 2022-12-05
> > **Response**
> >
> > Thank you for your clarifications.
> >
> > Just one note: if cycles are what you want to detect, then you can use persistent homology with any filter, the cycle will always appear as a point in the corresponding barcode (although its coordinates will depend on the filter used).

---

> > > ### Author Response · Authors · 2022-12-12
> > > **Response to Reviewer Qt5k**
> > >
> > > We thank the reviewer for the response, and an interesting note on persistent homology. We agree that one can use persistent homology with any filter to indicate that a cycle appears in the corresponding barcode, but it may not always be the case that non-homeomorphic cycles appear as different points in the corresponding barcodes.
> > >
> > > One of the examples we would like to refer the reviewer to are two graphs indicated in Figure 12 from the Appendix (which in fact are the same pairs of graphs indistinguishable by 1 and 2-WL tests or conventional GNNs). Let us suppose that one chooses a filtration function over the nodes and edges of the function as indicated in equation (70) in page 34 of the appendix. (The height corresponds to endowing constant height function over the set of nodes, and endowing weights on edges based on the Lp distance between features of two nodes connected by the edge.) Then one can check that the persistent barcodes represent the two non-homeomorphic cycles (one consisting of 4 nodes, and the other consisting of 3 nodes) as identical points.
> > >
> > > Previous studies have empirically shown that the choice of the filtration function for constructing persistent diagrams is crucial for detecting non-isomorphism classes of graphs. For example, a work by Horn et al. from “Topological Graph Neural Networks” (2022) demonstrate that static persistent homology falls short in distinguishing cycle structures of graphs in comparison to dynamic persistent homology, a technique which optimizes filtration function suitable for distinguishing a given set of graphs.
> > >
> > > The Cy2C-GNN architecture provides a different approach where one does not need to find suitable filtration function for each graph dataset, yet empirically performs on par to dynamic persistent homology in detecting cyclic substructures. These points are supported from ablation studies using the synthetic datasets proposed from previous literature, where one can observe that Cy2C-GNN can distinguish cycle structures more effectively than static persistent homology or conventional GCNs, and as effectively as dynamic persistent homology.

---

### Official Review · Reviewer_KiJD · 2022-10-25

**Confidence:** 3
**Correctness:** 3
**Technical Novelty And Significance:** 3
**Empirical Novelty And Significance:** 2
**Recommendation:** 5

**Clarity, Quality, Novelty And Reproducibility:**

As discussed above, the paper is generally not well-written and it is not very easy to read. The authors provide enough details for the reader to reproduce the reported results. The quality of the paper is good, but as discussed above there are several weak points. There is some novelty in the work and the method is of some interest.


**Strength And Weaknesses:**

Strengths:

- The proposed method is relatively simple, but still it is more powerful than several standard GNNs since it can distinguish graphs that WL and its variants cannot.

- The proposed approach seems to perform relatively well on the considered graph classification datasets. It outperforms all the baselines on 8 out of the 16 datasets.

Weaknesses:

- Since the authors claim that the Cy2C-GNN model can distinguish graphs that standard or even more powerful GNNs cannot distinguish, one would expect the authors to empirically verify that. There exist datasets that are created to measure the expressiveness of GNNs such as the Circular Skip Link dataset [1], and the EXP and CEXP datasets [2]. I would suggest the authors also evaluate the proposed model on these or other similar datasets.

- Even though the underlying idea is relatively simple, the paper is difficult to follow and contains mathematical notation that takes quite a bit of time to parse. I would thus recommend the authors to try and simplify the presentation.

- The main theoretical result (Theorem 4.3) shows that if one graph has a chordless cyclic element in its basis of different length than all the chordless cyclic elements of the basis of another graph, then the proposed method can distinguish the two graphs. Can all those graphs also be distinguished by some higher-order variant of WL (i.e., the $k$-WL algorithm for some $k>1$)? Are there any real-world problems which require such graphs to be mapped to different embeddings? What is also the intuition behind constructing a complete subgraph consisting of the nodes of a cyclic element?

- The authors discuss in details the computational complexity of the proposed method. However, they do not provide any empirical results. I would suggest the authors also measure the running time of the method (including preprocessing steps such as computing the set of cycles that form a basis for cycles of each input graph). Since the authors have evaluated Cy2C-GNN on REDDIT-M-5K, which contains large graphs, I would guess that the running time is not prohibitive.

- The writing is not very clear. There are several grammatical and syntactic mistakes, while notation is not always clearly defined (for instance, in Lemma 4.1, what do symbols $\cong$ and $\not \cong$ denote?). Therefore, the considered manuscript lacks clarity and I suggest the authors work on improving that.\
page 1: "The graph data set The adjacency matrix" --> The sentence does not make sense\
page 5: "to each nodes" --> "to each node"\
page 7: "3 social network dataset" --> "3 social network datasets"\
page 7: "3 small molecules" --> "3 small molecular datasets"

- Are there any results that link universal covers with stable colorings that emerge from the WL algorithm?

- Tables 1 and 2 have too small font size, and are not readable when printed. Please increase the font size.

[1] Murphy, R., Srinivasan, B., Rao, V., & Ribeiro, B., "Relational pooling for graph representations", In International Conference on Machine Learning, pp. 4663-4673, 2019.\
[2] Abboud, R., Ceylan, I. I., Grohe, M., & Lukasiewicz, T., "The Surprising Power of Graph Neural Networks with Random Node Initialization", In Proceedings of the Thirtieth International Joint Conference on Artificial Intelligence, pp. 2112-2118, 2021.


**Summary Of The Paper:**

The main contribution of this paper is Cy2C-GNN, a model that can distinguish pairs of isomorphic graphs that 1-, 2- and 3-WL tests cannot. The proposed approach identifies the cycle basis of the graph and then constructs complete subgraphs consisting of the nodes of each basis element. The clique adjacency matrix is created from these subgraphs. Then, the model produces some features by performing a neighborhood aggregation step using the clique adjacency matrix and these features are concatenated with features emerging from multiple standard neighborhood aggregation steps. The proposed model is evaluated on standard graph classification datasets where it achieves competitive performance with state-of-the-art methods.

**Summary Of The Review:**

In my view, the paper seems to be proposing an interesting contribution for the graph representation learning community. The quality of the paper is good, and a novel theoretical results is presented. The proposed method is simple and it seems to perform well on standard datasets. However, I have some concerns mainly with the clarity of the paper, the motivation behind some parts of the method, the significance of the theoretical result, and the empirical evaluation.

---

> ### Author Response · Authors · 2022-11-14
> **Response to Reviewer KiJD**
>
> We sincerely thank the reviewer for giving constructive and thorough feedback, which are of great help for improving the paper.
>
> [Additional Empirical Experiments]
>
> We thank the reviewer for suggesting us relevant references. Among the datasets the reviewer recommended, we experimentally verified that Cy2C-GNN outperforms baseline GCNs in classifying Circular Skip Link dataset, see Appendix B in the updated paper for further details. The dataset was obtained from the respective github repository, and we compared the performances of Cy2C-GCN with 1 message passing layer to GCN with 1, 2, 3, 4, and 5 message passing layers, RP-GIN, and GIN from the reference. Given that the dataset is only divided into training and validation set, we apply the stopping criterion for only the training dataset. As indicated in the summary of the paper, Cy2C-GCN not only transforms cyclic subgraphs into complete subgraphs, but also masks other edges which do not lie inside the cyclic subgraphs. Hence, we can deduce that Cy2C-GNN effectively highlights the cyclic structures of graphs in comparison to baseline GCNs.
>
> As for the second reference the reviewer kindly suggested, we would like to note that Theorem 3.3 of the paper mathematically supports the validity of GNNs introduced in the second reference. Theorem 3.3 states that baseline GNNs cannot distinguish a pair of graphs if and only if their universal covers and the pullback of node attributes are isomorphic. Random initialization of node attributes can be understood as a procedure of changing the pullback of node attributes over universal covers of graphs, thereby allowing GNNs to distinguish a wider class of graphs compared to baseline GNNs.
>
> [Mathematical Exposition]
>
> As the reviewer suggested, we reorganized some of the rigorous mathematical formulations to the appendix of the paper and supplemented the rest of the mathematical arguments with additional explanations. Please refer to the revisions made in the updated draft, indicated in blue.
>
> [Role of Cyclic Structures]
>
> This is a great question. We would like to point out that Examples A.36 and Remarks A.37 in the appendix demonstrates that Cy2C-GNNs can distinguish classes of graphs that 1, 2, and 3-WL isomorphism tests cannot distinguish. Previous studies showed that 3-WL isomorphism tests cannot distinguish strongly regular graphs, some of which consists of classes of graphs whose numbers of chordless cyclic graphs are different. Because Cy2C-GNNs can distinguish graphs with varying chordless cyclic substructures, we can prove that Cy2C-GNNs can distinguish graphs that are not distinguishable by some higher-order variants of WL.
>
> Cyclic structures appear in a wide range of real-world graph datasets. For example, chemical molecules and protein structures such as Decalin and Bicyclopentyl are chemical structures with isomorphic universal covers (i.e. indistinguishable by WL tests) but have different cyclic substructures. As for social network datasets, differences in cyclic substructures reflect global differences in social relations among a group of network users, whose interactions cannot be effectively distinguished or detected from adjacent (or local) social interactions. Both theoretical and experimental results suggest that Cy2C-GNN enriches and widens the zoo of distinguishable classes of graphs compared to baseline GNNs.
>
> [Computational Complexity]
>
> In Appendix B, we added new experiments on comparing the running times of Cy2C-GCN and baseline GCNs. Here, we included the pre-processing steps required for implementing Cy2C-GCNs, such as constructing clique adjacency matrices and detecting cycle bases. Since the REDDIT-M-5K dataset has the highest average number of nodes and edges in our benchmark datasets, we selected the dataset to conduct these experiments. For a fair comparison, we evaluated the GPU’s running time of baseline GCNs and Cy2C-GCNs with 128 hidden dimensions in the training sequence. We perform five iterations of implementing GCNs and Cy2C-GCNs with identical hyper-parameters, and measure the average time spent for reaching the same number of epochs to analyze additional computational costs derived from the clique adjacency matrix. We are able to infer from additional experiments that Cy2C-GNN have comparative execution times to baseline GCNs while ensuring to encapsulate cyclic structures of graphs.
>
> [Relation to higher variants of WL]
>
> Theorem 3.3 generalizes the results proved by Krebs and Verbitsky, who proved the correspondence between isomorphism classes of universal covers and the classes of graphs distinguishable by WL kernel algorithm. Here, the isomorphism classes of universal covers are assumed to have node attributes which are all identical. In Theorem 3.3, we generalize the result to hold for any isomorphism classes of universal covers whose node attributes are endowed in accordance to those endowed over the baseline graph datasets.

---

> > ### Author Response · Authors · 2022-11-14
> > **Response to Reviewer KiJD (continued)**
> >
> > [Typo / Grammatical Corrections]
> >
> > As the reviewer kindly pointed out, to the best of our effort, we made corrections to any grammatical errors we were able to find in the manuscript and clarified mathematical notations.
> >
> > [Table fonts]
> >
> > As the reviewer suggested, we enlarged the fonts for Tables 1 and 2. We reorganized references for the benchmark graph neural networks used in comparison to Cy2C-GNNs in the results section of the manuscript.

---

### Author Response · Authors · 2022-11-14
**Response to Reviewers**

We would like to sincerely thank all the reviewers for providing constructive and helpful feedbacks. Please refer to the updated manuscript, where we wrote in blue texts to indicate which parts of the manuscript we revised during the review period.

Here are a few key changes made in the revised manuscript.
1. We paraphrased some of the technical mathematical formulations, and made references to relevant sections in Appendices A and B. We also made corrections on grammatical errors or obscurely written phrases.
2. We included additional experiments on comparing the performance of Cy2C-GNN and baseline GCNs on classifying "Circular Skip Link dataset", a synthetic dataset designed to assess the ability of GNNs in distinguishing various cyclic structures of graphs. We also performed experiments on comparing execution times for implementing Cy2C-GNN and baseline GCNs. Both additional experiments can be found on Appendix B.
3. For Tables 1 and 2 from Section 5 of the main manuscript, we included best results obtained from baseline GCN, GAT, and GINs with message passing layers ranging between 1 and 5.

Please let us know if there are any other questions, suggestions, or comments.

---

### Decision · Program_Chairs · 2023-01-20

**Decision:**

Accept: poster

**Justification For Why Not Higher Score:**

AC recommends the authors more completely address reviewer’s remarks on relations with higher-order variant of WL.

**Justification For Why Not Lower Score:**

The authors did a nice job in response such as adding additional experiments suggested by reviewers (e.g., on classifying "Circular Skip Link dataset"), and clarifying mathematical expositions. Overall, the reviewers’ attitudes are positive regarding the contribution of this paper.

**Metareview: Summary, Strengths And Weaknesses:**

This paper proposes a new graph neural network architecture termed Cycle to Clique Graph Neural Network (Cy2C-GNN). The main contribution of this paper is Cy2C-GNN, a model that can distinguish pairs of isomorphic graphs that 1-, 2- and 3-WL tests cannot. The authors characterize and process the graph cycles using universal covers, which is motivated by a characterization of the classes of graphs that usual graph neural nets can distinguish. More precisely, they provide a simple yet efficient procedure for turning cycles into cliques and for processing clique adjacency matrices with new graph neural network layer combined with standard aggregation schemes. The proposed model is evaluated on standard graph classification datasets where it achieves competitive performance with state-of-the-art methods. The authors did a nice job in response such as adding additional experiments suggested by reviewers (e.g., on classifying "Circular Skip Link dataset"), and clarifying mathematical expositions. Overall, the reviewers’ attitudes are positive regarding the contribution of this paper. AC recommends the authors more completely address reviewer’s remarks on relations with higher-order variant of WL.

**Note From Pc:**

if the above contains the word "oral" or "spotlight" please see: "oral" presentation means -> notable-top-5% and "spotlight" means -> notable-top-25%. As stated in our emails, we are disassociating presentation type from AC recommendations